# Ether-compatible sulfurized polyacrylonitrile cathode with excellent performance enabled by fast kinetics via selenium doping

Xin Chen[1,2], Linfeng Peng[1], Lihui Wang[1,2], Jiaqiang Yang[2], Zhangxiang Hao[2], Jingwei Xiang[2], Kai Yuan[2], Yunhui Huang[2], Bin Shan[2], Lixia Yuan[2] & Jia Xie [1]

Sulfurized polyacrylonitrile is suggested to contain $S_n$ ($n \leq 4$) and shows good electrochemical performance in carbonate electrolytes for lithium sulfur batteries. However inferior results in ether electrolytes suggest that high solubility of $Li_2S_n$ ($n \leq 4$) trumps the limited redox conversion, leading to dissolution and shuttling. Here, we introduce a small amount of selenium in sulfurized polyacrylonitrile to accelerate the redox conversion, delivering excellent performance in both carbonate and ether electrolytes, including high reversible capacity (1300 mA h g$^{-1}$ at 0.2 A g$^{-1}$), 84% active material utilization and high rate (capacity up to 900 mA h g$^{-1}$ at 10 A g$^{-1}$). These cathodes can undergo 800 cycles with nearly 100% Coulombic efficiency and ultralow 0.029% capacity decay per cycle. Polysulfide dissolution is successfully suppressed by enhanced reaction kinetics. This work demonstrates an ether compatible sulfur cathode involving intermediate $Li_2S_n$ ($n \leq 4$), attractive rate and cycling performance, and a promising solution towards applicable lithium-sulfur batteries.

[1] State Key Laboratory of Advanced Electromagnetic Engineering and Technology, School of Electrical and Electronic Engineering, Huazhong University of Science and Technology, 430074 Wuhan, China. [2] State Key Laboratory of Materials Processing and Die & Mould Technology, School of Materials Science and Engineering, Huazhong University of Science and Technology, 430074 Wuhan, China. Correspondence and requests for materials should be addressed to L.Y. (email: yuanlixia@hust.edu.cn) or to J.X. (email: xiejia@hust.edu.cn)

The increasing demand for advanced energy storage has attracted extensive research of lithium-ion batteries[1]. Lithium–sulfur(Li–S) batteries gain interest due to the high theoretical energy density of 2600 W h kg$^{-1}$ [2–4]. However, the insulating nature of sulfur and end-product $Li_2S$ make them electrochemically inert, and the theoretical level is difficult to achieve[5–8]. Carbon sulfur composites with efficient conductive networks can usually deliver high capacity with good reactivity[9–13]. Nevertheless, it is challenging to maintain high capacity because of the dissolution of the lithium polysulfide intermediates $Li_2S_n$ followed by the "shuttling effect", which lead to loss of active materials, low Coulombic efficiency and poor cycle life[14–17]. Interestingly, with different sulfur sources, a different set of $Li_2S_n$ ($n = 2–8$) will be generated in the redox reaction, in which the solubility and the redox conversion of $Li_2S_n$ play an important role in cathode performance[18–20]. For instance, when elemental sulfur is used, high order polysulfide intermediates ($Li_2S_n$ ($n > 4$)) with high solubility will be generated[21–23]. Since it is difficult to convert all soluble $Li_2S_n$ ($n > 4$) into insoluble $Li_2S_2$ or end-product $Li_2S$ without dissolution, it is very challenging to shut down the "shuttling effect" and obtain high Coulombic efficiency and satisfactory cycling performance[24,25]. On the other hand, if only intermediate ($Li_2S_2$), which is close to completely insoluble, is involved such as in the case of small sulfur molecule ($S_{2–4}$) captured in microporous carbon, the dissolution of the polysulfides can be avoided due to a "quasi solid state" reaction mechanism[26]. Similar strategy has been employed with elemental sulfur cathodes in a recent work by Nazar and coauthors, which shows good capacity with low electrolyte/sulfur ratio and minimum dissolution of polysulfides[27]. However, it is reasonable to expect that without soluble polysulfide intermediates, the reaction kinetics as well as the rate capability will be limited presumably due to the intrinsic "quasi solid state" reaction mechanism[28–30]. Thus to develop sulfur cathodes with good capacity, long life and high rate, it is necessary to involve soluble polysulfide intermediates but mitigate polysulfides dissolution at the same time.

Surprisingly, sulfurized polyacrylonitrile (SPAN), reported by Wang and co-workers[31], seems to be a promising cathode which exhibits good capacity, reasonable rate capability, nearly 100% Columbic efficiency, good cycling performance and presumably no polysulfides dissolution in carbonate electrolyte. Though the exact structure of SPAN is still not clear, it is generally believed that sulfur is chemically bonded to the pyrolyzed pyridine ring initially and during the chemical reaction, and Li2Sn (n ≤ 4) is involved in the redox reaction. Recently the performance of SPAN cathodes has been further improved by using modified versions, such as SPAN@MWCNT[32], SPAN@GNS[33], NiS2-SPAN[34], and highly ordered MSPAN[35] and carbonized SeS2-PAN[36]. Among them, SPAN prepared by Archer's group exhibited high capacity and durability for 1000 cycles, but only in carbonate-based electrolytes[37]. It has been suggested that sulfur exists as $-S_x^{2-}-$ ($2 \leq n \leq 4$) units in S@pPAN, and there are no soluble intermediates during the redox process of S@pPAN in carbonate-based electrolyte[38]. Nevertheless, SPAN cathodes usually work well in carbonate electrolyte but show poor results in ether electrolyte[37,39–41]. It is reasonable to speculate that in carbonate electrolyte, $Li_2S_n$ ($n \leq 4$) shows lower solubility and can react with carbonate to form a protective solid electrolyte interphase (SEI) to mitigate the dissolution problem[42]. But in ether electrolyte the solubility of $Li_2S_n$ ($n \leq 4$) is high enough to trump the limited redox conversion and leads to the dissolution and the shuttling effect[28,30], which suggests that an underlying slow redox conversion and slow kinetics problem still exists despite that ether-based electrolytes evidently have better compatibility with lithium metal anodes[43,44]. Thus, it is desirable to accelerate the redox conversion of $Li_2S_n$ ($n \leq 4$) and boost the kinetics in SPAN, which should mitigate polysulfides dissolution and lead to compatibility with both carbonate and ether electrolyte as well as high rate and long cycle life performance.

Similar with S, selenium (Se) lies in the same column with S in the periodic table and shows much better kinetics in high-rate Se-based or Se-doped composites[45–49]. Recent work by Qian et al. shows that a catalytic amount of Se doping in an elemental sulfur carbon composite leads to a tremendous rate-boosting effect[50]. Different from other rate accelerators such as conducting carbons[51], metal oxides[3], metal sulfides[23], and metal nitrides[52], Se can not only easily achieve uniform distribution at a molecular level through Se–S bonding, but also contribute capacity. However, such a concept has only been shown in elemental sulfur cathodes in which a large amount of conducting carbon is still used and the cathode capacity is limited.

Herein, we design $Se_xSPAN$ ($x = 0.06, 0.09, 0.14$) composites as the cathode with the intention of using a catalytic amount of Se as both a rate accelerator and a capacity contributor in a polymeric framework. By accelerating the redox transformation of the only low order intermediate $Li_2S_n$ ($n \leq 4$) into insoluble $Li_2S_2$ or $Li_2S$, the dissolution problem should be largely mitigated. Indeed the experiment results show that compared with traditional SPAN, $Se_{0.06}SPAN$ cathode delivers high reversible capacity, superior rate performance and long cycles in both ether and carbonate electrolytes. The high rate performance could be attributed to higher electronic conductivity and faster lithium ion diffusion by Se-doping, which successfully serves as both a capacity contributor and a rate promotor in sulfurized polyacrylonitrile cathodes. As a result, the dissolution of polysulfides is mitigated in sulfurized polyacrylonitrile cathodes. This work not only demonstrates an ether-compatible sulfurized polyacrylonitrile cathode enabled by fast kinetics, to the best of our knowledge, one of the best rate and cycling performance simultaneously achieved, but also shows a sulfur cathode involving soluble polysulfides without polysulfide dissolution and shuttling. Such an approach provides a promising solution towards practical lithium sulfur batteries.

## Results

**Synthesis and characterization**. The scanning electron microscopy (SEM) images show the morphology of pyrolyzed PAN (pPAN) and $Se_{0.06}SPAN$ composites (Fig. 1a, b). The pPAN synthesized under Ar atmosphere at 300 °C is composed of irregular particles with a size around 500 nm. The $Se_{0.06}SPAN$ composite prepared by annealing the mixture of $Se_xS$ and PAN (3:1 by weight) at the same method is consisted of round particles with a size around 200 nm. The morphology difference is attributed to the dehydrogenation reaction between PAN and $Se_xS$. $Se_xS$ can dehydrogenate PAN to generate a conductive framework by firstly forming stable heterocyclic rings which is further dehydrogenated and substituted by $Se_xS$ to form cross-linking $C–S–Se_x–S–C$ chains or ring structures. The transmission electron microscopy (TEM) images of $Se_{0.06}SPAN$ composites (Fig. 1c) show that original particles are in circular-shape with a size around 200 nm, and these particles aggregate into a bulky cluster, which is consistent with the SEM images. High resolution transmission electron microscopy (HRTEM) is used to explore the microstructure of the $Se_{0.06}SPAN$ composite. It can be clearly observed that $Se_{0.06}SPAN$ composite has a uniform structure. The energy-dispersive X-ray spectroscopy (EDS) elemental mapping images reveal that the carbon elemental mapping image overlaps with sulfur and selenium mapping images, suggesting the homogeneous distribution in the $Se_{0.06}SPAN$ composite (Fig. 1d–h). The content of Se and S in the $Se_{0.06}SPAN$ composite

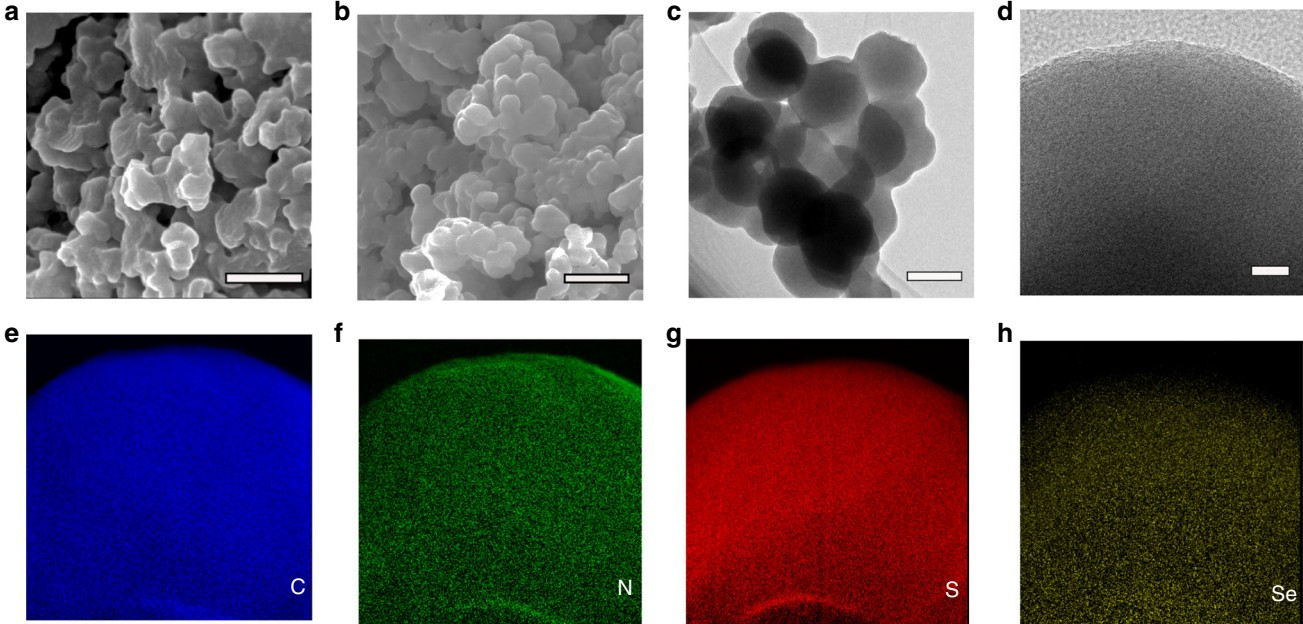

**Fig. 1** Materials characterization. SEM images of **a** pPAN. **b** $Se_xSPAN$ composites. **c** TEM image of $Se_xSPAN$ composite. **d** EDS elemental mapping images of the $Se_xSPAN$ composite, marked by a square, for carbon (**e**), nitrogen (**f**), sulfur (**g**), and selenium (**h**). Scale bars, 500 mm (**a**, **b**), 200 mm (**c**), 20 nm (**d**)

is 47.25% as shown in Supplementary Table 1, containing $Se_{0.09}SPAN$ and $Se_{0.14}SPAN$ composites with different doped proportions of Se. In the thermogravimetric analysis (TGA), the $Se_xSPAN$ composites show almost no weight loss below 400 °C (Supplementary Fig. 1).

The phase structure of $Se_{0.06}SPAN$ composites is explored by X-ray diffraction, Raman spectroscopy and FT-IR spectra. The XRD patterns of pPAN show one broad peak range from 17 to 26 degree (Fig. 2a). While pPAN shows a major peak at $2\theta = 17°$ corresponding to the (110) plane of the PAN crystalline structure, in the case of SPAN, such characteristic peak of pPAN disappears and a broad diffraction peak at $2\theta = 25°$ corresponding to the graphitic (002) plane newly presents after pyrolysis with sulfur[53]. The XRD patterns show quite different results between $Se_{0.029}S$ and $Se_{0.06}SPAN$. The latter shows an amorphous structure. The XRD peaks of $Se_{0.029}S$ are very similar with that of $Se_{0.06}S$ (Supplementary Fig. 6). Moreover, the XRD peaks of $Se_{0.06}S$ can not be observed in the XRD peaks of $Se_{0.06}SPAN$. It suggests that there is no $Se_{0.06}S$ in $Se_{0.06}SPAN$ composite. Although the proportions of Se are different, the $Se_{0.09}SPAN$ and $Se_{0.14}SPAN$ composites show similar patterns (Supplementary Fig. 3). The XRD patterns of $Se_{0.06}SPAN$ composite show one broad peak at 25 degree and no $Se_{0.029}S$ peak is detected, which suggests that the formation of an amorphous structure in $Se_{0.06}SPAN$ composite materials. It is noteworthy that the proportion of Se has been increased obviously from 0.029 to 0.06, which suggests that the dehydrogenation reaction is mainly caused by sulfur. The sorption analysis using Brunauer–Emmett–Teller (BET) theory (Fig. 2b) reveals a relatively low surface area of 18 $m^3 g^{-1}$.

To analyze the chemical bonds of the $Se_{0.06}SPAN$ composite, Raman spectra, Fourier-transform infrared spectroscopy (FT-IR) and X-ray photoelectron spectroscopy (XPS) are performed (Fig. 2c–f, Supplementary Fig. 4, 5). Specific peak assignments are summarized (Supplementary Fig. 4, 5 and Supplementary Tables 4, 5). As the previous reports[37], the peaks of SPAN located at 307 and 367 $cm^{-1}$ correspond to the C−S bonds and those at 470 and 926 $cm^{-1}$ correspond to the S−S stretch. However, the peak at 387 $cm^{-1}$ corresponding to the S−Se bond[50] appears in $Se_{0.06}SPAN$, instead of C−S bond. From the

results of Raman spectra, the structure of $Se_xSPAN$ is very similar with that of SPAN. Moreover, the Raman spectra provide information on the degree of crystallization of materials. It has been known that representative Raman spectra of carbon-based materials show two major peaks at 1325 and 1532 $cm^{-1}$, corresponding to the disordered D band and the graphitic G band, respectively. According to the intensity ratio ($I_G/I_D$) of these two bands, $Se_{0.06}SPAN$ shows a higher degree graphitization than that of SPAN, as the $I_G/I_D$ ratios for $Se_{0.06}SPAN$ and SPAN are 0.98 and 0.89, respectively. Meanwhile, the electronic conductivity of $Se_xSPAN$ more than doubled compared to SPAN through direct current (DC) polarization method (Supplementary Fig. 2 and Supplementary Table 3). On the basis of the information, the structure of the $Se_{0.06}SPAN$ can be confirmed to be a carbon construction via dehydrogenation and efficient π–π stacking, with sulfur covalently bonded to the carbon backbones simultaneously, which is also consistent with the XRD analysis.

Figure 2d shows the FT-IR spectra of SPAN and $Se_{0.06}SPAN$. It is noteworthy that the peaks are almost same between the two spectra. The peaks at 479 and 511 $cm^{-1}$ correspond to the S−S stretching, and the peak at 668 $cm^{-1}$ corresponds to the C−S stretching. Besides, the peak at 940 $cm^{-1}$ corresponds to the ring breathing in which a C−S bond is included. Therefore, these characteristic peaks verify the structure of C−S and S−S bonds after the dehydrogenation reaction. Moreover, the FT-IR analysis give information on the atom distribution of the carbon backbones. It has been known that the peaks at 1237 and 1498 $cm^{-1}$ and of $Se_{0.06}SPAN$ correspond to the symmetrical stretching of C = N and C = C bonds, respectively[37,53], and the peak at 801 $cm^{-1}$ corresponds to the ring breathing of six-membered rings[37,53]. Apparently, the structure of $Se_xSPAN$ composite is very similar with that of SPAN from the FT-IR analysis, which is consistent with the Raman analysis. These suggest that $Se_xSPAN$ also has complicated structure as well as SPAN.

As shown in Fig. 2e, there are two S2p peaks located at 163.6 and 164.9 eV, corresponding to the S−S and C−S bonds, respectively[34]. In addition, S2p peaks are also apparent at 161.8 and 168.1 eV. As shown in Fig. 2f, there are two S2p peaks located

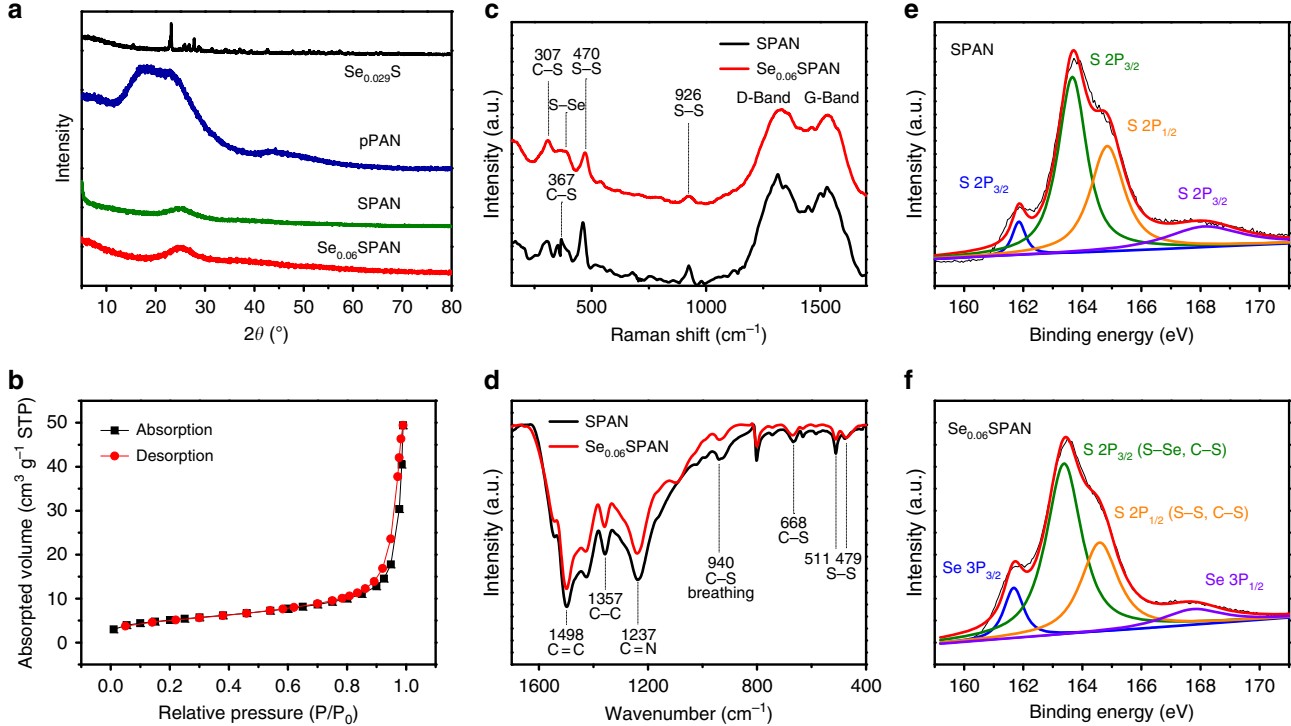

**Fig. 2** Materials characterization. **a** XRD patterns of $Se_{0.029}S$, pPAN, SPAN, and $Se_{0.06}SPAN$. **b** $N_2$ adsorption–desorption isotherms of $Se_{0.06}SPAN$. **c** Raman spectra **d** FTIR spectra of SPAN and $Se_{0.06}SPAN$. XPS analyses (S2p) of SPAN (**e**) and $Se_{0.06}SPAN$ (**f**) composites, respectively

at 163.3 and 164.6 eV, corresponding to the S−Se and S−S homopolar bonds, respectively[54,55]. The two S2p peaks also attribute to the C−S bond. Owing to the lower electron density of sulfur, the two binding energies at 163.4 and 164.6 eV represent a −0.2 and −0.3 eV shift for the S–Se and C–S bonds, respectively[54,55]. The two binding energies at 161.6 and 167.7 eV also represent a −0.2 and −0.4 eV shift, which also validat the interaction between Se and S in the $Se_{0.06}SPAN$[54,55]. According to the XPS spectrum (Supplementary Fig. 7a), the predominant peaks of C1s, N1s, S2p, and Se3d all exist. Meanwhile, the presence of Se−S bond is also confirmed by the Se3d XPS spectra. A doublet Se3d peak with binding energy of 55.6 and 56.4 eV is attributed to the S−Se bond, and this peak is curve-fitted (Supplementary Fig. 7b).

**Electrochemical performance of cathodes**. The rate performance of $Se_{0.06}SPAN$ composite cathodes is investigated in both ether-based and carbonate-based electrolytes (Fig. 3), with 1 mg cm$^{-2}$ active material loading. When cycling at a current density of 0.2 A g$^{-1}$ (0.13C) in ether-based electrolyte, the $Se_{0.06}SPAN$ composite cathodes deliver an initial capacity of 1680 mA h g$^{-1}$ based on S and Se. The $Se_{0.06}SPAN$ composite cathodes show significantly excellent rate performance in both ether and carbonate-based electrolytes. In ether-based electrolyte, they deliver a reversible capacity of 1320, 1210, 1160, 1110, 1030, 960, and 900 mA h g$^{-1}$ with the increasing current rate from 0.2 (0.13C), 0.4 (0.26C), 1 (0.65C), 2, 4, 6 to 10 A g$^{-1}$ (6.5C) respectively (Fig. 3a, b), corresponding to an excellent utilization ratio (84%) to its theoretical capacity (1546 mA h g$^{-1}$, Supplementary Table 2). The capacity then increases back to 1228 mA h g$^{-1}$ when the current density returns back to 0.4 A g$^{-1}$, demonstrating its superior stability to bear current changes. The reversible discharge capacities are slightly lower in the carbonate-based electrolyte, varying from 1173, 1115, 1081, 1048, 995, 957 to 847 mA h g$^{-1}$ with the

increasing current rate from 0.2, 0.4, 1, 2, 4, 6 to 10 A g$^{-1}$ respectively (Fig. 3c, d). When the current density returns to 0.4 A g$^{-1}$ after cycling at various rates, the discharge capacity of $Se_{0.06}SPAN$ composite cathode can recover to 1135 mA h g$^{-1}$. In contrast, the discharge capacities of traditional SPAN cathode decreases significantly at a varied current density in ether-based electrolyte (Supplementary Fig. 8b). These results indicate that the Se doping played a critical role for enhancing the rate performance by accelerating the reaction kinetics. It is clearly shown that the rate performance of the $Se_{0.06}SPAN$ composite cathode is higher than that of literature reports (Supplementary Fig. 13a and Supplementary Table 6).

The cycling performance of the cells with $Se_{0.06}SPAN$ composites cathode in ether-based electrolyte at a current density of 0.2 A g$^{-1}$ (0.13C), 1 A g$^{-1}$ (0.65C), 2 A g$^{-1}$ (1.3C) is shown in Fig. 4a. The reversible capacities of the cells are 1230, 1100, 1200 mA h g$^{-1}$ in the second cycle respectively. After 500 cycles, the cells with $Se_{0.06}SPAN$ composites cathode maintain a reversible capacity of 1130 mA h g$^{-1}$ at 0.2 A g$^{-1}$, possessing the capacity retention of 91.6% with almost 100% Coulombic efficiency based on the second discharge capacity. Apparently, it shows good compatibility with ether-based electrolytes. In contrast, the discharge capacity of traditional SPAN cathode decays significantly in ether-based electrolyte (Supplementary Fig. 8a). The phenomenon for traditional SPAN cathode in ether-based electrolyte can also be seen in other literature reports[37,42]. The $Se_{0.06}SPAN$ cathode is firstly cycled for 10 cycles with LiNO$_3$, and then cycled in ether-based electrolyte without LiNO$_3$. The cycle performance of $Se_{0.06}SPAN$ cathode is still good even without LiNO$_3$ (Supplementary Fig. 8c, d). It suggests that the superior electrochemical performance of $Se_{0.06}SPAN$ cathode is only caused by the introduction of catalytic Se. Thus, the LiNO$_3$ additive in ether-based electrolyte only has an effect on the protection of Li metal anode to achieve better electrochemical performance. The high reversible capacity, long cycle life and

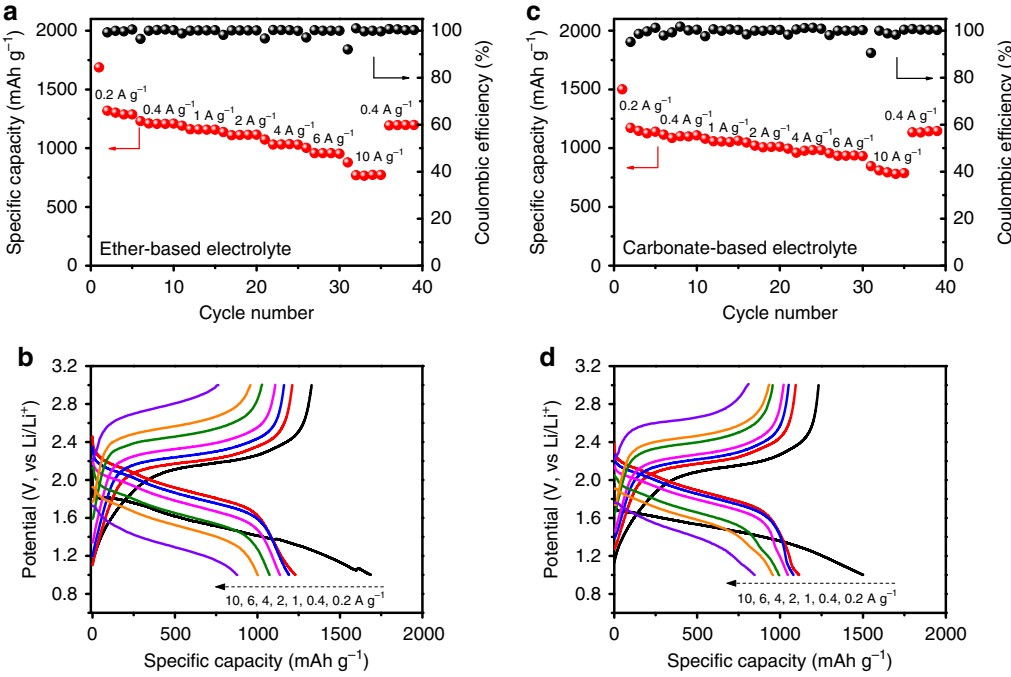

**Fig. 3** Electrochemical performance of the lithium–sulfur batteries. **a** Rate performance of the $Se_{0.06}SPAN$ measured at various current densities in ether-based electrolyte. The current densities are same for both charge and discharge in each cycle. **b** The corresponding electrochemical discharge and charge profiles of $Se_{0.06}SPAN$ at various cycles. **c** Rate performance of the $Se_{0.06}SPAN$ measured at various current densities in carbonate-based electrolyte. **d** The corresponding electrochemical discharge and charge profiles of $Se_{0.06}SPAN$ at various cycles

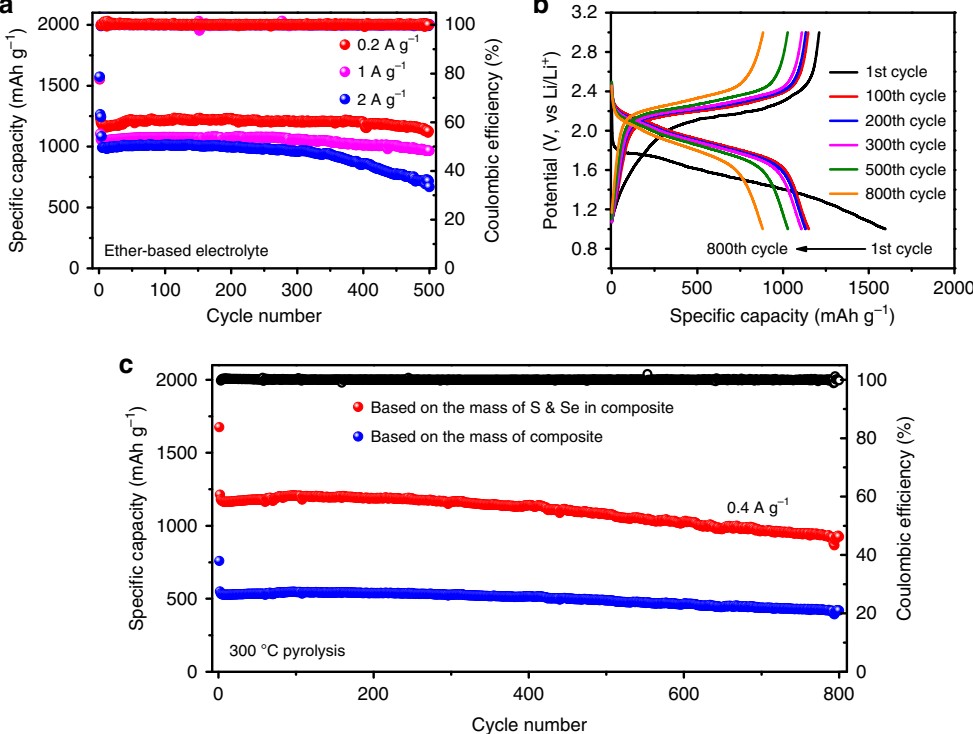

**Fig. 4** Electrochemical performance of the lithium–sulfur batteries. **a** Cycle performance of $Se_{0.06}SPAN$ at various current densities in ether-based electrolyte. **b** Electrochemical discharge and charge profiles of $Se_{0.06}SPAN$ at various cycles. The tests are performed at $0.4\,g^{-1}$. **c** Capacity and Coulombic efficiencies versus cycle number for $Se_{0.06}SPAN$. The red curve report capacities relative to the weight of the S and Se in the cathode, whereas the data represented by blue curve are the corresponding capacities based on the overall composite mass

high Coulombic efficiency of $Se_{0.06}SPAN$ composites cathodes demonstrate that the Se can accelerate redox kinetics and prevent the dissolution of the lithium polysulfides effectively. The reversible capacities of $Se_{0.06}SPAN$ composites cathode with 1 mg cm$^{-2}$ active material loading are 1150, 1131, 1107, 1028, and 881 mA h g$^{-1}$ based on S and Se after 100, 200, 300, 500, and 800 cycles with a current density of 0.4 A g$^{-1}$ (0.26C), respectively (Fig. 4b). The reversible capacity of the cell is 1156 mA h g$^{-1}$ based on S and Se in the second cycle (Fig. 4c). The capacity degradation upon repeated cycles of discharge and charge is only 0.029% per cycle from the second to 800th cycle, which hit a record among all reports cycled in ether-based electrolyte. Even up to 3 mg cm$^{-2}$ based on the mass of S and Se, the $Se_{0.06}SPAN$ composite cathodes still deliver a high specific capacity (Supplementary Fig. 9d). The $Se_{0.09}SPAN$ and $Se_{0.14}SPAN$ composite cathodes also show good cycle performance (Supplementary Fig. 9a, b) but slightly lower capacity due to higher proportion of Se (Supplementary Fig. 9c).

It is clearly shown that the cycle performance of $Se_{0.06}SPAN$ composite cathode is higher than the representative values of literature reports (Supplementary Fig. 13b and Supplementary Table 7). Based on the mass of the whole mass of $Se_{0.06}SPAN$, the cathode still delivers a capacity of 546 mA h g$^{-1}$ at second cycle, corresponding to an excellent utilization ratio (75%) to its theoretical capacity (726 mA h g$^{-1}$), and 416 mA h g$^{-1}$ even after 800 cycles based on the $Se_{0.06}SPAN$ composite (Fig. 4c). Compared to literature reports, the electrochemical performance of $Se_{0.06}SPAN$ composite cathode is at the top level. These results clearly demonstrate that the application of Se in $Se_{0.06}SPAN$ composite not only accelerate redox kinetics but also improve the cycle stability of the cells. In addition, the Coulombic efficiencies of the cells with $Se_{0.06}SPAN$ cathodes all remain at nearly 100% (except for the first cycle).

The cathode with 2.3 mg cm$^{-2}$ active material loading cycling in ether-based electrolyte is shown in Supplementary Fig. 10a. After 50 cycles, the cycled CR2025 coin cells in two different electrolytes are unpacked. The separator for the electrode cycled in ether-based electrolyte shows less color (Supplementary Fig. 11). It may be that the $Se_{0.06}SPAN$ composites hardly dissolve in ether-based electrolyte. Furthermore, the surface of Li metal from the cell cycling in carbonate-based electrolyte is darkened. The SEM image of Li metal is much smoother in ether-based electrolyte (Supplementary Fig. 12) than in carbonate-based electrolyte, which is an indication for better compatibility of ether-based electrolyte with Li metal.

## Discussion

The lithium ion transference is another factor to affect battery performance. The lithium ion diffusion coefficients ($D_{Li}^+$) for $Se_{0.06}SPAN$ are quantitatively calculated by a series of cyclic voltammetry (CV) with different scan rates. Randles–Sevick equation[56] is adopted, and then lithium ion diffusion coefficient is calculated based on the slop of the linear plot of the peak current ($I_p$) versus the square root of the scan rate ($v^{0.5}$). From the linear relationship of $I_p$ and $v^{0.5}$ (Supplementary Fig. 14), $D_{Li+}$ of two reduction peaks are obtained (Fig. 5a). It is worth noting that $D_{Li+}$ for the reduction and oxidation peaks of the Li–$Se_{0.06}SPAN$ battery are both higher than the Li–SPAN battery, suggesting that Se-doping facilitates fast Li-ion transport. This finding can be contributed to the catalytic effect of Se on the electrochemical performance for lithium storage, which is consistent of the superior rate performance and cycle performance of Li–$Se_xSPAN$ batteries. A comparison of reduction potentials is also studied to further analyze the effect of Se (Supplementary Fig. 15a). The reduction peaks for $Se_xSPAN$ electrode are 2.07 and 1.77 V, lower

than that of pure sulfur electrode (2.32 and 2.02 V), but higher than that of S/microC electrode (1.68 V). The two peaks at 2.32 and 2.02 V correspond to the reduction of $S_8$ to higher order polysulfides ($Li_2S_n$, $4 < n < 8$) and then to lower order polysulfides ($Li_2S_n$, $n \leq 4$)[30,57]. The 1.68 V peak is related to the reduction of smaller sulfur molecules confined within the microporous carbon (from $S_2$ to $Li_2S_2/Li_2S$)[28,30]. Because of the unique sulfur structure ($-S_x-$, $2 \leq x \leq 4$) in SPAN[38], the 2.07 and 1.77 V peaks correspond to the fast reduction of soluble $Li_2S_n$ ($n \leq 4$) and then to insoluble $Li_2S_2/Li_2S$. The typical reduction peaks of $Se_xSPAN$, pure sulfur, and S/microC are all in agreement with their own galvanostatic discharge curves (Supplementary Fig. 15b). The 1.8 V average plateau is observed for $Se_xSPAN$, which also falls in between elemental sulfur and small molecule sulfur cathodes. It also suggests that only soluble and lower order of polysulfides ($Li_2S_n$, $n \leq 4$) are involved. Thus the CV curves and voltage profiles suggest a reaction mechanism involving transition between soluble $Li_2S_n$ ($n \leq 4$) and insoluble $Li_2S_2/Li_2S$ during the lithiation and delithiation process. Owing to the fast Li-ion diffusion and reaction kinetics enabled by Se-doping, the soluble polysulfides can be transformed effectively, which inhibit polysulfides dissolution and the shuttling effect. Figure 5b shows discharge/charge voltage profiles of SPAN and $Se_{0.06}SPAN$ electrode at 0.2 A g$^{-1}$. The $Se_{0.06}SPAN$ electrode has a discharge capacity of 1240 mA h g$^{-1}$, much higher than that of SPAN electrode. Moreover, the $Se_{0.06}SPAN$ electrode possesses a relatively low polarization value of 0.42 V between the charge and discharge plateaus, which is much lower than that of 0.6 V for the SPAN. The improved discharge capacity and reductive polarization show that $Se_{0.06}SPAN$ is able to boost the electrochemical reaction kinetics during the discharge/charge processes in Li–S batteries.

To validate the above-mentioned points, we investigate the Li$^+$ migration within SPAN and $Se_xSPAN$ using density functional theory (DFT) model. The experimental data show that the migration energy barriers within SPAN and $Se_xSPAN$ are 1.33 and 0.39 eV, respectively (Supplementary Fig. 16). It is apparently that the structure of $Se_xSPAN$ is more beneficial for Li migration. The lower migration energy barrier is in agreement with the experimental results that the $D_{Li}^+$ of $Se_xSPAN$ is higher than that of SPAN. This finding likely explains that $Se_xSPAN$ has better reaction kinetics compared with SPAN. A lower barrier can lead to an increase in the diffusion rate according to the exponential rule. Faster Li$^+$ migration within the $Se_xSPAN$ can promote the redox reaction of soluble intermediates ($Li_2S_n$, $n \leq 4$), thus effectively to mitigate polysulfides dissolution and the shuttling effect.

The dissolution and diffusion of polysulfides during the redox process of SPAN and $Se_{0.06}SPAN$ cathodes are also investigated in ether-based electrolyte. The Li–SPAN and Li–$Se_{0.06}SPAN$ batteries after several cycles are both discharged to 2.0 V. Then, the SPAN and $Se_{0.06}SPAN$ cathodes are both rinsed with ether-based electrolyte after the batteries are unpacked. The solutions are measured by UV–Vis spectroscopy. As shown in Fig. 5c, for SPAN, the dissolution of polysulfides can be clearly detected in ether-based electrolyte in despite of undetectable polysulfides in carbonate-based electrolyte[37]. The SPAN for UV–Vis spectra test is prepared at 450 °C for 6 h by the same synthetic method according to the previous literature[37]. The sharp peak at ~280 nm corresponds to the $S_4^{2-}/S_6^{2-}$ species, the shoulder peak located at ~350 nm corresponds to the $S_4^{2-}$ species, and the peak at 250 nm to the $S_8^{2-}/S_6^{2-}$ species, in which $S_x^{2-}$ ($x = 4$–8) species are presumably generated from low order intermediates $S_n^{2-}$ ($n \leq 4$)[58,59]. Above results suggest that in ether-based electrolyte the dissolution and diffusion of polysulfides from the SPAN cathode occur, resulting in rapid capacity decay. However, polysulfides are undetectable for $Se_{0.06}SPAN$ cathode, suggesting that fast redox conversion of

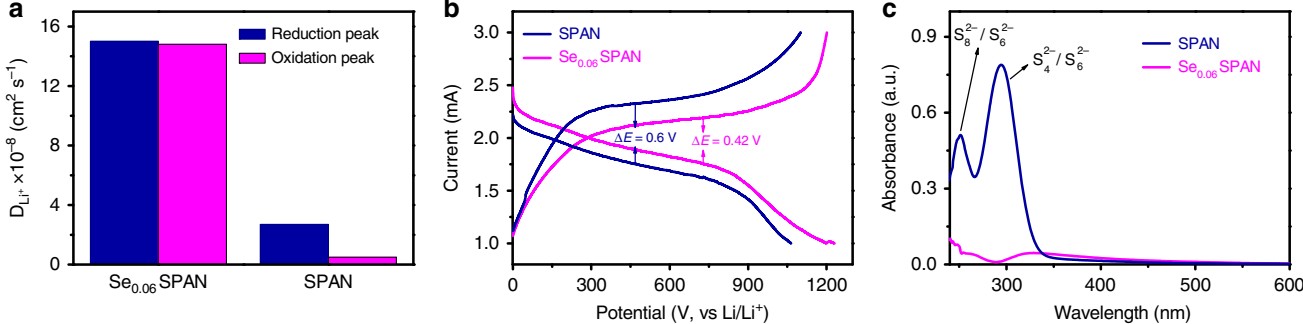

**Fig. 5** Improved electrochemical reaction kinetics. **a** $Li^+$ diffusion coefficients of reduction peak of $Se_{0.06}SPAN$ and SPAN cathodes. **b** Discharge–charge curves of SPAN and $Se_{0.06}SPAN$ electrodes. **c** UV–Vis absorption spectra of the solution after washing cycled SPAN and $Se_{0.06}SPAN$ cathodes

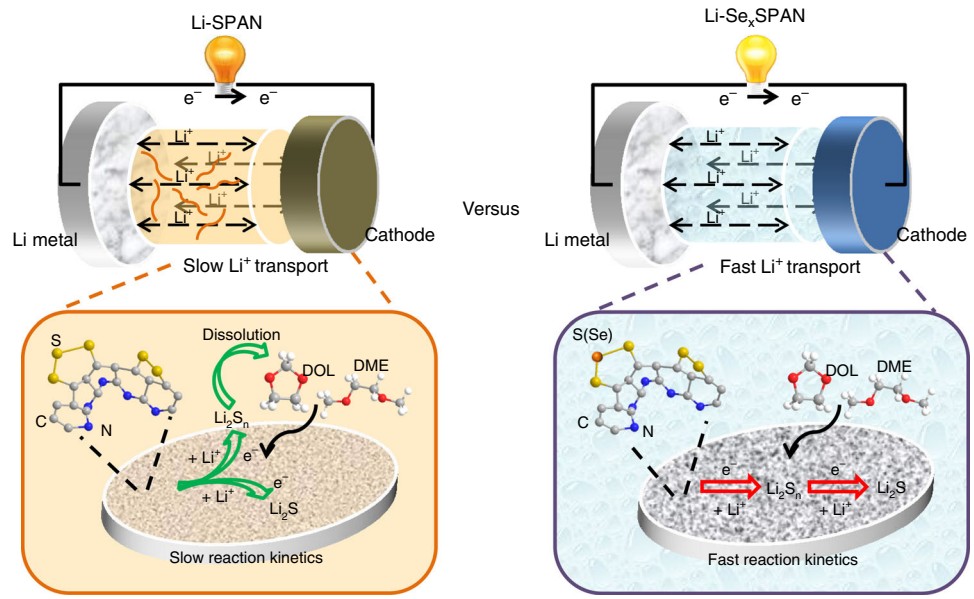

**Fig. 6** The scheme of proposed reaction process. Small amount of Se-doping significantly enhances the redox conversion of polysulfides and reaction kinetics, leading to ether-compatibility and superior performance of Li–$Se_{0.06}SPAN$ battery

polysulfide intermediates and fast kinetics of the $Se_{0.06}SPAN$ cathode in ether-based electrolyte. It is reasonable to conclude that the catalytic amount of Se-doping significantly enhances the redox conversion of polysulfides and reaction kinetics, thus leading to the compatibility with ether-based electrolyte and the outstanding electrochemical performance of Li–$Se_{0.06}SPAN$ batteries (Fig. 6). Similar rate boosting effects are seen in both Se and Tellurium (Te), which indicates the mechanism may be related to the promotion of electron transfer by Se and Te since Te is a well-known catalyst to promote electron transfer in organic chemistry[60]. Further studies of detailed mechanism and performance optimization are still ongoing.

In summary, we design a selenium-doped sulfurized polyacrylonitrile ($Se_xSPAN$, $x < 0.15$, ~50 wt% $Se_xS$) cathode which shows excellent electrochemical performance and outstanding compatibility with ether-based electrolyte. The catalytic amount of Se-doping leads to higher lithium ion diffusion coefficient, relatively low polarization, resulting in rapid conversion of polysulfide intermediates and fast reaction kinetics, which in turn prevent the dissolution of polysulfide intermediates in ether. It is believed that the solution by rapid and complete conversion from soluble $Li_2S_n$ ($n \leq 4$) to insoluble $Li_2S$ can also be applied in other similar cathodes involving $Li_2S_n$ ($n \leq 4$). The solution should also be universal for all sulfur cathodes involving $Li_2S_n$ ($n \leq 4$). This work shows a sulfur cathode involving intermediate $Li_2S_n$ ($n \leq 4$) with excellent compatibility with ether-based electrolyte, simultaneously achieved rate and cycling performances that are among the best for sulfur cathodes, to the best of our knowledge, and a promising solution towards applicable lithium sulfur batteries.

## Methods

**Synthesis of selenium-sulfur composites**. Commercial sulfur and selenium powders are ball-milled at a weight ratio of 15:1, 12:1, and 10:1 with ethanol as the dispersant. The resulting mixture is dried at 60 °C for 6 h in a vacuum oven. Then the mixture is fully filled in a 200 mL Teflon-lined stainless steel autoclave, heated at 250 °C for 12 h and then cooled to room temperature naturally. The mole and weight ratio of sulfur and selenium in $Se_xS$ will reduce slightly during the preparation compared to the initial mixing ratio.

**Synthesis of selenium-doped sulfurized polyacrylonitrile**. A $Se_xS$ and PAN mixture is milled uniformly at a weight ratio of 3:1 for 0.5 h in a mortar and then annealed at 300 °C for 2.5 h in argon atmosphere. During this step, slight excess amount of $Se_xS$ is used. A part of Se and S will be removed from the reactant either in S or Se gas form or $H_2S$ gas form generated in dehydrogenation reaction at 300 °C. Since S is easier to evaporate than Se, the resulting S:Se ratio will be lower than the initial mixing ratio. For the PAN/S composites without selenium, the preparation procedure is similar, except the adding selenium.

**Material characterization**. X-ray powder diffractometer (PANalytical X'pert PRO-DY2198, Holland) is used to collect X-ray diffraction (XRD) patterns, operating at 40 kV and 40 mA using Cu Ka radiation ($\lambda = 0.15406$ nm) and determining structure refinement of the XRD patterns using GSAS software. The

test of Brunauer–Emmett–Teller (BET) surface area is measured at −196 °C by a BET analyzer (Micromeritics ASAP 2010, US). Scanning electron microscopy (SEM) (FEI, Sirion 200) is used to collect images of pPAN and $Se_xSPAN$ samples. Transmission electron microscopy (TEM) (FEI, Tecnai G200) is used to collect images of $Se_xSPAN$ samples. Raman spectra are measured by LabRAM HR800 (Horiba Jobin Yvon). Fourier Transform Infrared Spectra are taken using a Bruker Vertex 70 FTIR spectrometer. The X-ray photoelectron spectroscopy (XPS) analysis of $Se_xSPAN$ and SPAN materials is conducted with a Kratos Analytical spectrometer (AXIS-ULTRA DLD-600W) and an Al Kα (1486.6 eV) X-ray source, and the binding energy values are calibrated using the C 1s peak at 285.0 eV. The thermal gravimetric (TG) profiles are determined by thermal gravimetric analysis (PerkinElmer) in an argon atmosphere with a heat rate of 10 °C min$^{-1}$ from 25 to 1000 °C. The UV–Vis spectrum are characterized by UV-2550 spectrophotometer (SHIMADZU). To reveal the electronic conductivities of SPAN and $Se_xSPAN$, symmetric cells with configurations of Au/SPAN/Au and Au/$Se_xSPAN$/Au are tested by Solartron 1470E CellTest system.

**Electrochemical measurements**. All of the electrochemical measurements are tested on 2032 coin cells with lithium foil as the anode which assemble under an argon-filled glove box ($H_2O$, $O_2 < 1$ ppm). A working electrode is prepared by mixing $Se_xSPAN$ composites, super P, sodium carboxyl methyl cellulose (NaCMC) and styrene butadiene rubber (SBR) at a weight ratio of 80:10:5:5. The slurry is coated onto Al foil and then dried in a vacuum oven at 70 °C for 12 h. The active material density of each cell is determined to be 1–3 mg cm$^{-2}$. A metallic Li sheet is used as the counter electrode, and 1 M LiPF$_6$ in a mixture of ethylene carbonate/dimethylcarbonate (EC/DMC; 1:1 by volume) as the electrolyte (Zhuhai Smooth way Electronic Materials Co., Ltd (China)). The amount of electrolyte is about 60 μL in each cell during assembling. The ether electrolyte is 1 mol L$^{-1}$ LiTFSI in a mixed solution of 1,2-dimethoxyethane (DME) and 1,3-dioxolane (DOL) (1:1 v/v) with LiNO$_3$ (2 wt%) as the additive. Electrolyte is added to each coin cell at the amount of 30 μL in each cell during assembling. The electrochemical workstation (CHI614b) is used to test cyclic voltammetry (CV) profiles at a scan rate of 0.05 mV s$^{-1}$ from 1 to 3 V at room temperature. The battery measurement system (Land, China) is used to measure the electrochemical performance and profiles from 1 to 3 V.

**Lithium ion diffusion coefficient**. Lithium ion diffusion coefficients for $Se_xSPAN$ are calculated by a series of cyclic voltammograms at different scan rates, and the peak current data are analyzed with Randles-Sevick equation as following: $I_P = 2.69 \times 10^5 \, n^{1.5} \, A D_{Li}^+ {}^{0.5} \, C_{Li} \, v^{0.5}$ in which $D_{Li}^+$ represented lithium ion diffusion coefficient (cm$^2$ s$^{-1}$), $I_P$ stood for the peak current in ampere (A), $n$ is the number of electrons involved in the reaction ($n = 2$ for Li–S battery), $A$ is the geometric area of the active electrode (cm$^2$), $C_{Li}$ referred to the lithium ion concentration (mol L$^{-1}$) and $v$ represented the scanning rate (V s$^{-1}$).

**DFT calculations**. The migration energy barriers reported herein were calculated using DFT within Perdew-Burke-Ernzerhof (PBE) generalized gradient approximation (GGA) via Vienna Ab initio Simulation Package (VASP). Projector augmented wave (PAW) method is used to depict the electron-ion interactions. The cutoff energy of 500 eV and Monkhorst−Pack k-meshes of $3 \times 1 \times 1$ is set for the calculations. Vacuum layers of at least 10 Å is needed for non-periodic directions. The Gaussian broadening with a width of 0.05 eV is used for the integration of the first Brillouin zone. The structure is optimized until the Hellmann-Feynman force is smaller than 0.05 eV Å$^{-1}$. Climbing-image nudged elastic band (CI-NEB) method is used to find the Li migration energy barriers.

## Data availability
The data that support the findings of this study are available from the corresponding authors upon request.

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

## Acknowledgements

This work was supported by the National Basic Research Program of China (973 Program, 2015CB258400), the National Science Foundation of China (Grant Nos. 21773077 and 51532005) and the Program for HUST Interdisciplinary Innovation Team (2015ZDTD021). The authors gratefully acknowledge the Analytical and Testing Center of HUST for allowing us to use its facilities.

## Author contributions

X.C. and L.F.P. contributed equally to this work. J.X., L.X.Y. and X.C. contributed the idea, experiments designed and manuscript writing. X.C. and L.F.P. carried out the experiments and the material characterizations. L.H.W. contributed the material synthesis. J.Q.Y. and B.S. contributed the theoretical calculations. Z.X.H., J.W.X., K.Y. and Y.H.H. contributed the discussion and manuscript editing. All authors contributed the comments on the final manuscript.

## Additional information

**Competing interests:** The authors declare no competing interests.

**Journal peer review information**: Nature Communications thanks Jou-Hyeon Ahn for their contribution to the peer review work. Peer reviewer reports are available.

