## [Peer Review File · Nature Communications]

Reviewers' comments:

Reviewer #1 (Remarks to the Author):

The authors have designed and synthesized a selenium doped sulfurized polyacrylonitrile, (SexSPAN, $x < 0.15$, ~50wt% SexS) as an ether compatible composite in the present study, though the formation of the compound and the mechanism of electrochemical lithiation-delithiation remains ambiguous.

Comments regarding the manuscript are listed below,

1. The methods section details that "Commercial selenium and sulfur powders were ball-milled at a weight ratio of $15:1$, $12:1$ and $10:1$ with ethanol as the dispersant..." Primarily how do the authors end up with Se in doping concentrations?
2. The formation of SexSPAN composite is unclear as per the references 33 , and 48 (*Carbonized Polyacrylonitrile-Stabilized Se_x Cathodes for Long Cycle Life and High Power Density Lithium Ion Batteries. Adv. Funct. Mater. 24, 4082-4089, 449, 2014, Selenium-Doped Cathodes for Lithium-Organosulfur Batteries with Greatly Improved Volumetric Capacity and Coulombic Efficiency, Adv. Mater. 29, 1701294, 2017*).
3. The survey spectrum and Se3d spectrum of the prepared SexSPAN composite need to be analyzed.
4. Please include the conductivity studies supporting the fast kinetics and reaction boosting with the claimed Se doping.
5. The authors have utilized LiNO₃ additive for this study with which it cannot make sure whether the obtained stable electrochemical performances are attributed to the claimed Se doping.
6. On what basis is the theoretical capacity of the composite calculated? Elaborate the same and include in the manuscript.
7. Computational studies to validate the obtained results are encouraged.

Reviewer #2 (Remarks to the Author):

This manuscript reports Se doped SPAN cathode materials which demonstrate stable cycling performances almost no polysulfide dissolution and shuttle in ether electrolyte. After minor revisions, this manuscript is suitable for Nat. Common.

What is the specific capacity calculation, based on S or SeS? Please show the specific capacity based on whole composite materials including pPAN.

Se doping effectively suppress the polysulfide dissolving. How about the electrochemical performances in DOL+DME without LiNO₃ additive?

Authors proposed a schematic structure of SPAN as shown in supplementary Fig. 12. If there are C-S bonds, the theoretical capacity of sulfur or SeS will be lower than 1672 mAh/g of elemental sulfur. Please provide evidences to support the schematic structure. If no evidence, suggest to delete it.

Reviewer #3 (Remarks to the Author):

The scientific works presented are original. S-Se composites are already widely reported in the literature as well as PAN/S composite electrodes, while PAN/S-Se has not yet been reported. The results reported show high capacities, and very good capacity retention even taking into account the total mass of the electrode with a capacity of 500 mAh/g. The addition of a small amount of Se in sulfur, seems to have a large impact on the reaction kinetic. However some points are not well explain and some of them are confusing. The end of the introduction is similar than the abstract, with the incorporation of results. The introduction must present the state of the art, pose the problematic and highlight the originality of the work and not give results.

The comparison of the XRD data of samples Se_{0.029}S and Se_{0.06}SPAN is not relevant, since the SeS alloys don't have the same composition, the XRD study of an alloy Se_{0.06}S must be carried out to

demonstrate the alloy amorphization in the presence of PAN.

For the determination of the Li^+ diffusion, the parameters aren't well defined, the surface A , can not be the geometric area of the electrode, as the electrode is a porous one, the determination of real surface area must be done, furthermore why $n= 2$? several peaks were observed on the CV (Figure 11, it's a complex reaction with several intermediates, the electrochemical reaction associated with the reaction involved at I_p red must be given, and the number of electron involved in the reaction must be commented. In the same manner, on the scheme 1 the electrochemical reactions are not balance.

In the paper, the higher diffusion coefficient of Li^+ is associated with the catalytic effect of Se. The Li^+ diffusion is related to mass transport whereas Se seems to have an effect on the transfer reaction at the interface. Some explanations must be given to explain clearly the link between the catalytic effect of Se on the charge transfer kinetics (electron) and the diffusion of lithium ions in the electrolyte.

Why the amount of electrolyte is not the same for carbonate based electrolyte (60 μL) and for the ether one (30 μL), as the PS dissolution is largely associated with its concentration, the comparison is only pertinente with the same amount of electrolyte.

The potential of the cell is very low (the average potential is only 1.8V), compared to the classical Li/S cell, some comments on the mechanism must be given in order to explain the potential shape.

Itemized list of response to the editor' and reviewers' remarks

(Black italic: Reviewer's remarks; Blue type: Our response)

Reviewer(s)' Comments to Author:

Reviewer #1:

The authors have designed and synthesized a selenium doped sulfurized polyacrylonitrile, (Se_xSPAN , $x < 0.15$, $\sim 50\text{wt}\%$ Se_xS) as an ether compatible composite in the present study, though the formation of the compound and the mechanism of electrochemical lithiation-delithiation remains ambiguous.

Reply to the Reviewer: Thanks the referee for the professional review work on our manuscript. From your valuable comments, there are several points that need to be addressed. According to your nice suggestions, we have made extensive corrections. The detailed corrections are listed below.

1. The methods section details that "Commercial selenium and sulfur powders were ball-milled at a weight ratio of $< b > 15:1$, $12:1$ and $10:1 < /b >$ with ethanol as the dispersant..." Primarily how do the authors end up with Se in doping concentrations?

Reply to the Reviewer: Thanks the referee for the valuable comments. The Se doping concentration is dictated by the initial S : Se mixing ratio. The mole and weight ratio of sulfur and selenium in the final product will reduce during the preparation compared to the initial mixing (*Energy & Environmental Science*, 2015, 8, 1754-5692). For example, in the first step, S powder is sealed with Se powder (weight ratio S : Se = 15 : 1, mole ratio S : Se = 1 : 0.027) in a stainless seal vessel under argon atmosphere protection, then heated at 250 °C for 12 h, which resulted in $Se_{0.029}S$ composite (weight ratio S : Se = 14 : 1, mole ratio S : Se = 1 : 0.029). The lower weight ratio than initial mixing ratio is because that sulfur is much easier to evaporate and a small portion of S didn't get into Se_xS alloy (*Advanced Materials*, 2015, 27, 569-575).

In the second step, $Se_{0.029}S$ powder is mixed with commercial PAN powder (3 : 1, w/w) and heated at 300 °C for 2.5 h in a quartz tube furnace under argon atmosphere, which resulted in $Se_{0.06}SPAN$ (mole ratio S : Se = 1 : 0.06, weight ratio 6.75 : 1) composite. During this step, excess amount of $Se_{0.029}S$ is used. Large amount of Se and S be removed from the reactant either in S or Se gas form or H_2S gas form generated in dehydrogenation reaction at 300 °C. Since S is easier to evaporate than Se, the resulting S : Se ratio will be reduced. The preparation method and the results are quite common and consistent with previous studies in literature. These two steps are highly repeatable. By using the same reaction conditions, the same Se doping concentration can be obtained. Thus by using different mixing ratio of S : Se and controlling reaction conditions, different Se doping concentration can be achieved.

The experiment section of synthesis of Se_xS and Se_xSPAN composite has been revised with more detailed in manuscript as listed in the following.

Page 18: “Then the mixture is fully filled in a 200 mL Teflon-lined stainless steel autoclave, heated at 250 °C for 12 h and then cooled to room temperature naturally. The mole and weight ratio of sulfur and selenium in Se_xS will reduce slightly during the preparation compared to the initial mixing ratio.”

Page 18: “A PAN and Se_xS mixture is milled uniformly at a weight ratio of 3:1 for 0.5 h in a mortar and then annealed at 300 °C for 2.5 h in argon atmosphere. During this step, excess amount of Se_xS is used. Large amount of Se and S will be removed from the reactant either in S or Se gas form or H_2S gas form generated in dehydrogenation reaction at 300 °C. Since S is easier to evaporate than Se, the resulting S : Se ratio will be reduced.”

2. The formation of Se_xSPAN composite is unclear as per the references 33, and 48 (*Carbonized Polyacrylonitrile-Stabilized SeS_x Cathodes for Long Cycle Life and High Power Density Lithium Ion Batteries. Adv. Funct. Mater. 24, 4082-4089, 449, 2014, Selenium-Doped Cathodes for Lithium–Organosulfur Batteries with Greatly Improved Volumetric Capacity and Coulombic Efficiency, Adv. Mater. 29, 1701294, 2017*).

Reply to the Reviewer: Thanks the referee for the thoughtful comments. Firstly, the Se_xSPAN composite is very different from the materials reported in the two mentioned references. The structure of Se_xSPAN (heat-treated at 300 °C) composite is very similar with the SPAN structure from our characterization but with very small amount of Se doping. SPAN has been studied extensively and is well known for the complicated structure. Recently, it has been demonstrated that in the SPAN material, the sulfur exists as $-\text{S}_x-$ ($2 \leq x \leq 4$) units and chemically bonded with sulfurized PAN backbone (Energy Storage Materials, 2018, 14, 272-278). On the other hand, in the work of “*Carbonized Polyacrylonitrile-Stabilized SeS_x Cathodes for Long Cycle Life and High Power Density Lithium Ion Batteries. Adv. Funct. Mater. 24, 4082-4089*”, the structure is very different from traditional SPAN. SeS_2 is used as the starting material and the preparation is done in much higher carbonization temperature (600 °C), thus listed as the CPAN material by the authors. The resulting material $\text{SeS}_{0.7}/\text{CPAN}$ show Se : S ratio of 1 : 0.7 with more Se than S and fully carbonized polyacrylonitrile backbone with very weak C-S bond. This material is in a way similar to a Se/carbon composite material but with quite a lot of sulfur. Using large amount of Se will improve volumetric energy density but lower gravimetric energy density. Meanwhile, the electrochemical test was only done in carbonate electrolyte without showing ether compatibility.

In the work of “*Selenium-Doped Cathodes for Lithium–Organosulfur Batteries with Greatly Improved Volumetric Capacity and Coulombic Efficiency, Adv. Mater. 29, 1701294*”, the Se : S mole ratio is around 1 : 4 and the Se doping concentration is still

much higher than that in our Se_{0.06}SPAN (Se : S mole ratio 1 :16.7). Meanwhile the polymer backbone is formed by polymerization of the olefin group in the allyl moiety. Thus the polymer backbone is similar to the polypropylene which has very poor electroconductivity. But in our Se_{0.06}SPAN materials, the polymer backbone is suggested to be polypyridine which have aromatic pyridine rings with good electroconductivity (*Journal of the American Chemical Society*, 2015, 137, 0002-7863). In fact, presumably due to the different polymer backbone, the rate performance of SeS₂/PDATt cathode with much higher concentration of Se is still inferior than that of our Se_{0.06}SPAN. Moreover, same rational as in comparing with the first work, higher concentration of Se will decrease the specific capacity of the composite due to the low theoretical specific capacity of Se (675 mAh g⁻¹). Therefore, it is desirable to improve the electrochemical performance of the composite through the introduction of catalytic amount of Se. Furthermore, compared to traditional SPAN, the introduction of catalytic amount of Se largely boosts the electron transfer and accelerate the reaction kinetics, resulting in almost no polysulfides dissolution and shuttling effect. The fast kinetics enable ether compatibility which can't be achieved in traditional SPAN material.

To further differ from the two cathode with relatively simple structure in references, the description of similarities between Se_{0.06}SPAN and SPAN has been add to be more complete in manuscript.

Page 8-9: “However, the peak of SPAN at 387 cm⁻¹ corresponded to the S–Se bond⁴⁵ appeared in Se_{0.06}SPAN, instead of C–S bond. From the results of Raman spectra, the structure of Se_xSPAN is very similar with that of SPAN. Moreover... crystallization of materials.”

Page 9-10: “It had been known that the peaks at 1237 cm⁻¹ and 1498 cm⁻¹ and of Se_{0.06}SPAN corresponded to the symmetrical stretching of C=N and C=C bonds, respectively,^{34, 46} and the peak at 801 cm⁻¹ corresponded to the ring breathing of six-membered rings.^{34, 46} Apparently, the structure of Se_xSPAN composite is very similar with that of SPAN from the FT-IR analysis, which is consistent with the Raman analysis. These suggest that Se_xSPAN also has complicated structure as well as SPAN.”

3. The survey spectrum and Se3d spectrum of the prepared Se_xSPAN composite need to be analyzed.

Reply to the Reviewer: Thanks the referee for the valuable suggestions. Owing the S-Se bond can be shown in the S 2p spectra, we did not show the Se 3d spectra. As suggested, the XPS spectrums have been added in the Support Information Figure 5. According to the XPS spectrum (Supplementary Fig. 7a), the predominant peaks of C1s, N1s, S2p and Se3d all exist. The presence of Se–S bonds is also confirmed by the Se3d XPS spectra. A doublet Se3d peak with binding energy of 55.6 and 56.4 eV is

attributed to the S–Se bond, and this peak is curve-fitted (Supplementary Fig. 7b).

Supplementary Figure 7. a) XPS spectra, b) Se3d of the $\text{Se}_{0.06}\text{SPAN}$ composite.

Experimental data of XPS spectrum has been included in Supplementary Figure 6 and discussed in manuscript.

Page 10: “The two binding energies at 161.6 and 167.7 eV also represented a -0.2 and -0.4 eV shift, which also validated the interaction between Se and S in the $\text{Se}_{0.06}\text{SPAN}$.^{47, 48} According to the XPS spectrum (Supplementary Fig. 7a), the predominant peaks of C1s, N1s, S2p and Se3d all exist. The presence of Se–S bonds is also confirmed by the Se3d XPS spectra. A doublet Se3d peak with binding energy of 55.6 and 56.4 eV is attributed to the S–Se bond, and this peak is curve-fitted (Supplementary Fig. 7b).”

4. Please include the conductivity studies supporting the fast kinetics and reaction boosting with the claimed Se doping.

Reply to the Reviewer: Thanks the referee for the valuable suggestions. To reveal the electronic conductivities of SPAN and Se_xSPAN , we tried the four-probe method which failed to give any readings, presumably due to the low conductivity (similar as in a recent report *Energy Storage Mater.* 2018, 194). Then we tried direct current (DC) polarization method in symmetric cells with configurations of Au/SPAN/Au and Au/ Se_xSPAN /Au, which are done by using Solartron 1470E CellTest system. (as reported in “High-performance all-solid-state Li–Se batteries induced by sulfide electrolytes”, *Energy & Environmental Science*, 2018, online). The testing results are listed in Supplementary Table 3 and show that the conductivity of Se_xSPAN more than doubled compared to SPAN, which suggests that the introduction of catalytic amount of Se can lead to higher conductivity and presumably accelerate the reaction kinetics.

Supplementary Table 3. Electric conductivity results of SPAN and $\text{Se}_{0.06}\text{SPAN}$ using direct current (DC) polarization method.

Material	Resistance	Length (cm)	Area (cm ²)	Electric conductivity (S/cm)
SPAN	4.7×10^7	0.089	0.785	2.4×10^{-9}
Se _{0.06} SPAN	1.8×10^7	0.083	0.785	5.8×10^{-9}

Supplementary Figure 2. Equilibrium current of the Se_{0.06}SPAN and SPAN at different set voltages.

The electronic conductivity of Se_xSPAN and SPAN has been tested through direct current (DC) polarization method. The data are now clearly stated and added in manuscript, Supplementary Figure 2 and Table 3.

Page 8: “According to the intensity ratio (I_G/I_D) of these two bands, Se_{0.06}SPAN shows a higher degree graphitization than that of SPAN, as the I_G/I_D ratios for Se_{0.06}SPAN and SPAN are 0.98 and 0.89, respectively. Meanwhile, the electronic conductivity of Se_xSPAN more than doubled compared to SPAN through direct current (DC) polarization method (Supplementary Table 3 and Fig. 2). On the basis of the information, the structure of the Se_{0.06}SPAN can be confirmed to be a carbon construction *via* dehydrogenation and efficient π - π stacking,.....”

Page 19: “The UV-vis spectrum are characterized by UV-2550 spectrophotometer (SHIMADZU). To reveal the electronic conductivities of SPAN and Se_xSPAN, symmetric cells with configurations of Au/SPAN/Au and Au/Se_xSPAN/Au are tested by Solartron 1470E CellTest system.”

5. The authors have utilized LiNO₃ additive for this study with which it cannot make sure whether the obtained stable electrochemical performances are attributed to the claimed Se doping.

Reply to the Reviewer: Thanks the referee for the valuable suggestions. From the control experiments as shown in the following figure, the traditional SPAN cathode exhibits rapid decay of specific capacity in ether-based electrolyte with or without LiNO₃ additive (For clear comparison, we didn't add the results without LiNO₃ before.). However, the Se_xSPAN cathode using the ether-based electrolyte with LiNO₃ deliver

excellent cycle performance in Figure. 4. Without LiNO_3 Se_xSPAN cathode still can cycle more than 50 cycles using the ether-based electrolyte. The control experiment suggests that Se doping is the key difference in enabling ether compatibility. The LiNO_3 additive in ether based electrolyte is suggested to form stable solid electrolyte interface (SEI) on the Li metal anode (*Nature Communication, 2015, 6, 7436*). The stable SEI on the Li metal guarantee high coulombic efficiency and better anode cycling, which allow better evaluation of sulfur cathode performance. Otherwise, we have to replace lithium metal anode after 50-100 cycles for better evaluation of cathode performance. The collective results in previous work from the literature and our control experiments suggest that traditional SPAN is not compatible with ether-based electrolyte with or without LiNO_3 due to the rapid dissolution of polysulfide intermediates in ether-based electrolyte. Especially, the traditional SPAN cathode exhibits severe shuttle effect and a few cycles in ether-based electrolyte without LiNO_3 additive (*Journal of the American Chemical Society, 2015, 137, 12143*). The SPAN cathode exhibits slow reaction kinetics and the slow transition from soluble Li_2S_n ($n \leq 4$) to insoluble $\text{Li}_2\text{S}_2/\text{Li}_2\text{S}$, which resulting in the severe polysulfides dissolution and shuttling effect. But the Se_xSPAN cathode with the same electrolyte exhibits fast reaction kinetics and the fast transition from soluble Li_2S_n ($n \leq 4$) to insoluble $\text{Li}_2\text{S}_2/\text{Li}_2\text{S}$ enabled by the catalytic Se, which resulting in the almost no polysulfides dissolution and shuttle. It has been demonstrated that the introduction of catalytic Se has a positive influence on the stable electrochemical performances.

We modified Supplementary Figure 8a and added the cycling results of the Se_xSPAN cathodes without LiNO_3 and SPAN cathode with LiNO_3 in ether-based electrolyte for comparison.

Supplementary Figure 8. a) Cycle performance of the Se_xSPAN cathodes using ether-based electrolyte with and without LiNO_3 , and SPAN cathode with LiNO_3 in ether-based electrolyte. b) Rate performance of SPAN in the ether-based electrolyte.

Figure S6 in the literature shows that the traditional SPAN cathode exhibits severe shuttle effect and only a few cycles in ether-based electrolyte without LiNO_3 additive despite of the excellent cycle performance in carbonate-based electrolyte (*Journal of the American Chemical Society*, 2015, 137, 12143).

The cycle performance of Se_xSPAN cathode using ether-based electrolyte without LiNO_3 has been added in Supplementary Figure 8 and discussed in manuscript.

Page 12: “The phenomenon for traditional SPAN cathode in ether-based electrolyte could also be seen in other literature reports.^{34, 39} The Se_xSPAN cathode still can cycle more than 50 cycles using the ether-based electrolyte without LiNO_3 (Supplementary Figure 8a). The control experiment suggests that Se doping is the key difference in enabling ether compatibility.”

6. On what basis is the theoretical capacity of the composite calculated? Elaborate the same and include in the manuscript.

Reply to the Reviewer: The theoretical capacity is based on the Se and S together because Se and S contribute to the specific capacity during electrochemical reaction. For example, the theoretical capacity of $\text{Se}_{0.06}\text{SPAN}$ cathode based on Se and S is 1546 mAh g^{-1} . Since the mole ratio of Se : S is 0.06 : 1, the mass fraction of Se and S is 0.13 and 0.87 respectively. Owing to the theoretical capacity of Se (678 mAh g^{-1}) and S (1675 mAh g^{-1}), the theoretical capacity based on Se and S can be calculated. The theoretical capacity of composite based on whole composite material including PAN can be calculated due to the $\text{Se}_{0.06}\text{S}$ content (47%) of $\text{Se}_{0.06}\text{SPAN}$.

The theoretical capacity of the composite based on Se and S is now clearly stated and added in manuscript and Supplementary Table 2. The theoretical capacity of composite based on whole composite material including PAN is also added.

Page 11: “In ether-based electrolyte, it delivered a reversible capacity of 1320, 1210, 1160, 1110, 1030, 960 and 880 mAh g^{-1} with the increasing current rate from 0.2 (0.13 C), 0.4 (0.26 C), 1 (0.65 C), 2, 4, 6 to 10 A g^{-1} (6.5 C) respectively (Fig. 3a, 3b),

corresponding to an excellent utilization ratio (84%) to its theoretical capacity (1546 mAh g⁻¹, Supplementary Table 2). ”

Page 13: “The figure also showed that based on the mass of the composite, Se_{0.06}SPAN composites delivered a capacity of 546 mAh g⁻¹ at second cycle, corresponding to an excellent utilization ratio (75%) to its theoretical capacity (726 mAh g⁻¹), and 416 mAh g⁻¹ even after 800 cycles based on the Se_{0.06}SPAN composite. ”

Supplementary Table 2. Theoretical capacity of Se_xSPAN composite.

Materials	Se (wt%) /S (wt%)	S&Se (wt%) in composite	Theoretical capacity (mAh g ⁻¹ , based on Se and S)	Theoretical capacity (mAh g ⁻¹ , based on the composite)
Se _{0.06} SPAN	13/87	47.25	1546	726
Se _{0.09} SPAN	19/81	47.85	1485	710
Se _{0.14} SPAN	26/74	48.77	1415	690

7. Computational studies to validate the obtained results are encouraged.

Reply to the Reviewer: Thank the referee for the constructive comments. Computational studies indeed will be very helpful and provide more insights to our obtained results. Currently, there are several computational studies available for some basic understanding of reversible Se-S bond during cycle, which resulting in good electrochemical performance (*Advanced Materials*, 2017, 29, 1701294; *ACS Nano*, 2016, 10, 8289-8298). Although the detailed structures of SPAN and Se_xSPAN are extremely complex, we try our best to build a simplified model molecules to explore the Li ion diffusion barrier for SPAN and Se_xSPAN using density functional theory (DFT) calculation via VASP code. As a result, we find that the diffusion barrier for Se_xSPAN is indeed smaller than that for SPAN. This result is consistent with the experimental results showing that D_{Li⁺} of Se_xSPAN is higher than that of SPAN. This finding likely explains that Se_xSPAN has better reaction kinetics compared to SPAN. A lower barrier can lead to an increase in the diffusion rate according to the exponential rule, and faster diffusion on the Se_xSPAN can promote the redox reaction of soluble intermediates (Li₂S_n, n≤4).

We have added the results of DFT calculation in Supplementary Figure 16 and discussed in manuscript. The fast reaction kinetic of Se_xSPAN enabled by Se-doping has been explained more clearly. Hence, the soluble polysulfides can be transformed effectively, which inhibit polysulfides dissolution and the shuttling effect.

Page 15-16: To validate the above-mentioned points, we simulate the diffusion barriers for Li ion on SPAN and Se_xSPAN using density functional theory (DFT) calculation via VASP code. The energy profiles along the diffusion coordinate for Se_xSPAN is 0.39 eV, which is smaller than the diffusion barrier for SPAN, which is

1.33 eV according to our simulations (Supplementary Fig. 16). This result is consistent with the experimental results showing that D_{Li^+} of Se_xSPAN is higher than that of SPAN. This finding likely explain that Se_xSPAN have better reaction kinetics compared with SPAN. A lower barrier can lead to an increase in the diffusion rate according to the exponential rule. Faster lithium diffusion on the Se_xSPAN can promote the redox reaction of soluble intermediates (Li_2S_n , $n \leq 4$), thus effectively to mitigate polysulfides dissolution and the shuttling effect.

Page 20: **DFT calculations.** We perform the density functional theory (DFT) calculation via VASP code. The exchange and correlation energy is in the form of generalized gradient approximation (GGA) with Perdew–Burke–Ernzerhof (PBE). Projector augmented wave (PAW) method is used to depict the electron-ion interactions. The cutoff energy of 500 eV and Monkhorst–Pack k-meshes of $3 \times 1 \times 1$ are set for the calculations. Vacuum layers of at least 10 Å is needed for non-periodic directions. The Gaussian broadening with a width of 0.05 eV is used for the integration of the first Brillouin zone. The structure is optimized until the Hellmann–Feynman force is smaller than 0.05 eV/Å. Climbing-image nudged elastic band (CI-NEB) method is used to find the Li diffusion barriers.

Supplementary Figure 16. a) SPAN b) Se_xSPAN model for Li ion diffusion barriers employing DFT calculation; c) Energy profiles for diffusion processes of Li ion on SPAN and Se_xSPAN ; Schematic representations of corresponding diffusion pathway for SPAN from d) to e); Schematic representations of corresponding diffusion pathway for Se_xSPAN from f) to g).

Reviewer #2:

This manuscript reports Se doped SPAN cathode materials which demonstrate stable cycling performances almost no polysulfide dissolution and shuttle in ether electrolyte. After minor revisions, this manuscript is suitable for Nat. Commun.

Reply to the Reviewer: We would like to thank you first for all constructive comments of our manuscript. From your comments, there are several points that need to be addressed. According to your nice suggestions, we have made extensive corrections. The detailed corrections are listed below.

1. What is the specific capacity calculation, based on S or SeS? Please show the specific capacity based on whole composite materials including pPAN.

Reply to the Reviewer: The theoretical capacity is based on the Se and S together because Se and S contribute to the specific capacity during electrochemical reaction. For example, the theoretical capacity of $\text{Se}_{0.06}\text{SPAN}$ cathode based on Se and S is 1546 mAh g^{-1} . Since the mole ratio of Se : S is 0.06 : 1, the mass fraction of Se and S is 0.13 and 0.87 respectively. Owing to the theoretical capacity of Se (678 mAh g^{-1}) and S (1675 mAh g^{-1}), the theoretical capacity based on Se and S can be calculated. The theoretical capacity of composite based on whole composite material including PAN can be calculated due to the $\text{Se}_{0.06}\text{S}$ content (47%) of $\text{Se}_{0.06}\text{SPAN}$.

The theoretical capacity of the composite based on Se and S is now clearly stated and added in manuscript and Supplementary Table 2. The theoretical capacity of composite based on whole composite material including PAN is also added.

Page 11: “In ether-based electrolyte, it delivered a reversible capacity of 1320, 1210, 1160, 1110, 1030, 960 and 880 mAh g^{-1} with the increasing current rate from 0.2 (0.13 C), 0.4 (0.26 C), 1 (0.65 C), 2, 4, 6 to 10 A g^{-1} (6.5 C) respectively (Fig. 3a, 3b), corresponding to an excellent utilization ratio (84%) to its theoretical capacity (1546 mAh g^{-1} , Supplementary Table 2).”

Page 13: “The figure also showed that based on the mass of the composite, $\text{Se}_{0.06}\text{SPAN}$ composites delivered a capacity of 546 mAh g^{-1} at second cycle, corresponding to an excellent utilization ratio (75%) to its theoretical capacity (726 mAh g^{-1}), and 416 mAh g^{-1} even after 800 cycles based on the $\text{Se}_{0.06}\text{SPAN}$ composite.”

2. Se doping effectively suppress the polysulfide dissolving. How about the electrochemical performances in DOL+DME without LiNO_3 additive?

Reply to the Reviewer: Thanks the the referee for the valuable suggestions. From the control experiments as shown in the following figure, the traditional SPAN cathode exhibits rapid decay of specific capacity in ether-based electrolyte with or without

LiNO₃ additive (For clear comparison, we didn't add the results without LiNO₃ before). However, the Se_xSPAN cathode using the ether-based electrolyte with LiNO₃ deliver excellent cycle performance in Figure. 4. Without LiNO₃ Se_xSPAN cathode still can cycle more than 50 cycles using the ether-based electrolyte. The control experiment suggests that Se doping is the key difference in enabling ether compatibility. The LiNO₃ additive in ether based electrolyte is suggested to form stable solid electrolyte interface (SEI) on the Li metal anode (*Nature Communication, 2015, 6, 7436*). The stable SEI on the Li metal guarantee high coulombic efficiency and better anode cycling, which allow better evaluation of sulfur cathode performance. Otherwise, we have to replace lithium metal anode after 50-100 cycles in order to evaluation. The collective results in previous work from the literature and our control experiments suggest that traditional SPAN is not compatible with ether-based electrolyte with or without LiNO₃ due to the rapid dissolution of polysulfide intermediates in ether-based electrolyte. Especially, the traditional SPAN cathode exhibits severe shuttle effect and a few cycles in ether-based electrolyte without LiNO₃ additive (*Journal of the American Chemical Society, 2015, 137, 0002-7863*). The SPAN cathode exhibits slow reaction kinetics and the slow transition from soluble Li₂S_n (n≤4) to insoluble Li₂S₂/Li₂S, which resulting in the severe polysulfides dissolution and shuttling effect. But the Se_xSPAN cathode with the same electrolyte exhibits fast reaction kinetics and the fast transition from soluble Li₂S_n (n≤4) to insoluble Li₂S₂/Li₂S enabled by the catalytic Se, which resulting in the almost no polysulfides dissolution and shuttle. It has been demonstrated that the introduction of catalytic Se has a positive influence on the stable electrochemical performances.

We modified Supplementary Figure 8a and added the cycling results of the Se_xSPAN cathodes without LiNO₃ and SPAN cathode with LiNO₃ in ether-based electrolyte for comparison.

Supplementary Figure 8. a) Cycle performance of the Se_xSPAN cathodes using ether-based electrolyte with and without LiNO₃, and SPAN cathode with LiNO₃ in ether-based electrolyte. b) Rate performance of SPAN in the ether-based electrolyte.

Figure S6 in the literature shows that the traditional SPAN cathode exhibits severe shuttle effect and only a few cycles in ether-based electrolyte without LiNO_3 additive despite of the excellent cycle performance in carbonate-based electrolyte (*Journal of the American Chemical Society*, 2015, 137, 12143).

The cycle performance of Se_xSPAN cathode using ether-based electrolyte without LiNO_3 has been added in Supplementary Figure 7 and discussed in manuscript.

Page 11: “The phenomenon for traditional SPAN cathode in ether-based electrolyte could also be seen in other literature reports.^{34, 39} The Se_xSPAN cathode still can cycle more than 50 cycles using the ether-based electrolyte without LiNO_3 (Supplementary Figure 8a). The control experiment suggests that Se doping is the key difference in enabling ether compatibility.”

3. Authors proposed a schematic structure of SPAN as shown in supplementary Fig. 12. If there are C-S bonds, the theoretical capacity of sulfur or SeS will be lower than 1672 mAh/g of elemental sulfur. Please provide evidences to support the schematic structure. If no evidence, suggest to delete it.

Reply to the Reviewer: Thank the referee for the valuable comments. As suggested, we have deleted the schematic structure of SPAN.

Reviewer #3

The scientific works presented are original. S-Se composites are already widely reported in the literature as well as PAN/S composite electrodes, while PAN/S-Se has not yet been reported. The results reported show high capacities, and very good capacity retention even taking into account the total mass of the electrode with a capacity of 500 mAh/g. The addition of a small amount of Se in sulfur, seems to have a large impact on the reaction kinetic.

Reply to the Reviewer: Thank the referee for the valuable positive comments. From your comments, there are several points that need to be addressed. According to your nice suggestions, we have made extensive corrections. The detailed corrections are listed below.

1. *The end of the introduction is similar than the abstract, with the incorporation of results. The introduction must present the state of the art, pose the problematic and highlight the originality of the work and not give results.*

Reply to the Reviewer: Thank the referee for the valuable comments. We have modified the introduction to present the state of the art more clearly. The problematic issues have been added in the manuscript. The originality has also been highlighted. Furthermore, the results of experiment have been simplified. The end of the introduction has been modified to differ from the abstract.

Page 3-4: “On the other hand, if only intermediate (Li_2S_2) which is close to completely insoluble is involved such as in the case of small sulfur molecule (S_n ($n \leq 4$)) captured in microporous carbon, the dissolution of the polysulfides can be avoided due to a “quasi solid state” reaction mechanism.²⁵ Similar strategy has been employed with elemental sulfur cathodes in a recent work by Nazar and coauthors, which shows good capacity with low electrolyte/sulfur ratio and minimum dissolution of polysulfides.²⁶ However, it is reasonable to expect that without soluble polysulfide intermediates, the reaction kinetics as well as the rate capability will be limited presumably due to the intrinsic “quasi solid state” reaction mechanism.²⁷⁻²⁹. Thus to develop sulfur cathodes with good capacity, long life and high rate, it is necessary to involve soluble polysulfides intermediates but mitigate polysulfides dissolution at the same time. Surprisingly, sulfurized polyacrylonitrile (SPAN), firstly reported by Wang and co-workers in 2002, seems to be a promising cathode which exhibits good capacity, reasonable rate capability, nearly 100% Columbic efficiency, good cycling performance and presumably no polysulfides dissolution, however only in carbonate electrolyte.³⁰ Though the exact structure of SPAN is still not clear, it is generally believed that sulfur initially is chemical bonded to the pyrolyzed pyridine ring and during the chemical reaction and Li_2S_n ($n \leq 4$) is involved in the redox reaction.”

Page 5-6 “Thus, it is desirable to accelerate the redox conversion of Li_2S_n ($n \leq 4$) and boost the kinetics in SPAN, which should mitigate polysulfides dissolution and lead to compatibility with both carbonate and ether electrolyte as well as high rate and long cycle life performance.

Similar with S, selenium (Se) lies in the same column with sulfur in the periodic table and shows much better kinetics in high rate Se-based or Se-doping composites.⁴⁴⁻⁴⁶ Recent work by Qian et al shows that a catalytic amount of Se doping in elemental sulfur carbon composite led to tremendous rate boosting effect.⁴⁷ Different from other rate accelerators such as conducting carbons,⁴⁸ metal oxides,³ metal sulfides²² and metal nitrides,⁴⁹ selenium can not only can easily achieve uniform distribution at molecular level through Se-S bonding, but also contribute capacity. However, such concept has only been shown in elemental sulfur cathode in which a large amount of conducting carbon is still used and the cathode capacity is limited. Herein, we design Se_xSPAN ($x \approx 0.06, 0.09, 0.14$) composites as the cathode with the intention to use a catalytic amount of Se as both rate accelerator and capacity contributor in a polymeric framework. By accelerating the redox transformation of the only low order intermediate Li_2S_n ($n \leq 4$) into insoluble Li_2S_2 or Li_2S , the dissolution problem should be largely mitigated. Indeed the experiment results show that compared with traditional SPAN, $\text{Se}_{0.06}\text{SPAN}$ cathode delivers high reversible capacity, superior rate performance and long cycles in both ether and carbonate electrolytes. The high rate performance could be attributed to higher electronic conductivity and faster lithium ion diffusion by Se-doping which successfully serves as both capacity contributor and rate promotor in sulfurized polyacrylonitrile cathodes. As the result, the dissolution of polysulfides is mitigated in sulfurized polyacrylonitrile cathodes. To the best of our knowledge, this work not only demonstrates an ether-compatible sulfurized polyacrylonitrile cathode enabled by fast kinetics with one of the best rate and cycling performance achieved at the same time, but also shows the sulfur cathode involving soluble polysulfides without polysulfide dissolution and shuttling. Such approach should provide a promising solution towards applicable lithium sulfur batteries.”

2. *The comparison of the XRD data of samples $\text{Se}_{0.029}\text{S}$ and $\text{Se}_{0.06}\text{SPAN}$ is not relevant, since the SeS alloys don't have the same composition, the XRD study of an alloy $\text{Se}_{0.06}\text{S}$ must be carried out to demonstrate the alloy amorphization in the presence of PAN.*

Reply to the Reviewer: The XRD peaks of $\text{Se}_{0.029}\text{S}$ are very similar with that of $\text{Se}_{0.06}\text{S}$. Moreover, the XRD peaks of $\text{Se}_{0.06}\text{S}$ can not be observed in the XRD peaks of $\text{Se}_{0.06}\text{SPAN}$. It suggests that there is no $\text{Se}_{0.06}\text{S}$ alloy in $\text{Se}_{0.06}\text{SPAN}$ composite. Fourier transform infrared (FTIR) spectra indicates the formation of heterocyclic compounds. Raman spectra are also determine the C-S bond in the composite. The presence of C-S bond is further confirmed by the S 2p XPS spectra.

Supplementary Figure 6. XRD patterns of $\text{Se}_{0.029}\text{S}$ and $\text{Se}_{0.06}\text{S}$ composites.

Experimental data of $\text{Se}_{0.029}\text{S}$ and $\text{Se}_{0.06}\text{S}$ composites analyzed by XRD has been add in Supplementary Figure 6 and stated in the manuscript to make this point more clearly.

Page 8: “The XRD pattern showed quite different results between $\text{Se}_{0.029}\text{S}$ and $\text{Se}_{0.06}\text{SPAN}$. The latter showed an amorphous structure. The XRD peaks of $\text{Se}_{0.029}\text{S}$ are very similar with that of $\text{Se}_{0.06}\text{S}$ (Supplementary Fig. 6). Moreover, the XRD peaks of $\text{Se}_{0.06}\text{S}$ can not be observed in the XRD peaks of $\text{Se}_{0.06}\text{SPAN}$. It suggests that there is no $\text{Se}_{0.06}\text{S}$ alloy in $\text{Se}_{0.06}\text{SPAN}$ composite.”

3. For the determination of the Li^+ diffusion, the parameters aren't well defined, the surface A , can not be the geometric area of the electrode, as the electrode is a porous one, the determination of real surface area must be done, furthermore why $n=2$? several peaks were observed on the CV (Figure 11, it's a complex reaction with several intermediates, the electrochemical reaction associated with the reaction involved at I_p red must be given, and the number of electron involved in the reaction must be commented. In the same manner, on the scheme 1 the electrochemical reactions are not balance.

Reply to the Reviewer: Thanks the review for the valuable comments. For the solid-phase lithiation-delithiation reaction, in most cases, the diffusion of Li^+ within the solid electrode is the control step due to the slow process. Thus, the Li^+ diffusion rate usually determines the reaction rate, and the chemical diffusion coefficient of Li^+ can imply the reaction rate constant. Generally, the higher the chemical diffusion coefficient of Li^+ in lithium battery, the better the rate capability and the power density (*Journal of Power Sources*, 2005, 139, 261-268). So, the Li -ion diffusion coefficient is one of the most important fundamental parameters for the electrode material.

But, the determination of the diffusion coefficient is not easy. The process of Li^+ diffusion within the solid phase is influenced by many factors, including the concentration gradient, chemical potential and so on. The diffusion coefficient can be measured by the CV (Cyclic Voltammetry), GITT (Galvanostatic Intermittent Titration

Technique), PITT (Potentiostatic Intermittent Titration Technique), EIS (Electrochemical Impedance SPECTROSCOPY). Whatever the technology, the surface area is required. But the determination of the real surface is difficult due to the complexity of real electrochemical reaction and the volume expansion of electrode. Different measuring means always obtained different results. The specific surface determined via the most widely used BET method is not the surface that participate the electrode reaction. Here the Li^+ diffusion coefficient based on the CV results is not an absolute value. The calculated value is just relatively accurate and comparable due to the same synthesis and analysis condition of Se_xSPAN and SPAN composite. The synthesis and analysis of the Se_xSPAN and SPAN samples are strictly controlled under the same conditions. And the BET results of the Se_xSPAN and SPAN sample are very close. So we can infer that the Se_xSPAN and SPAN cathodes have the same electrochemical reaction area. So, the comparison result makes sense although the Li^+ diffusion coefficient calculated based on the geometric area is only the apparent value.

In addition, so far, the complex structure of SPAN (Se_xSPAN) is still not be completely determined and the related electrochemical reaction process is not fully understood either. Consequently, the number of electron involved in the reaction cannot be totally determined. It is accepted that the reaction of Li- Se_xSPAN battery involves multiple reaction, including the reaction from the $-\text{S}_x-$ ($2 \leq x \leq 4$) unites to soluble Li_2S_n ($n \leq 4$) and from soluble Li_2S_n ($n \leq 4$) to insoluble $\text{Li}_2\text{S}_2/\text{Li}_2\text{S}$, similar to the sulfur cathode based on elemental sulfur. So, most of the reported researches regarding SPAN adopt the same treatment with the Li-S system that using the total number of the electron in the reaction to determine the apparent diffusion coefficient (*Angewandte Chemie International Edition, 2014, 53, 10099-10104; Proceedings of the National Academy of Sciences, 2017, 114, 840-845*). Anyway, the application of the geometric area and the total number of the electron involved in the reaction should not affect the relative value of the diffusion coefficient for the Se_xSPAN and SPAN system although they are not the absolute value. It is reasonable that the higher Li^+ diffusion coefficient of Li- Se_xSPAN battery suggests that catalytic Se largely boosts the electron transfer and accelerate the reaction kinetics.

At last, the scheme 1 is the scheme of the reaction mechanisms of the Li/ Se_xSPAN and Li/SPAN cells. The balanced equation is not given due to the complex structure and electrochemical reaction process of Li/ Se_xSPAN and Li/SPAN. So far, the complex structure of SPAN (Se_xSPAN) has not been completely determined and the related electrochemical reaction process is not fully understood either. The scheme for the electrode reaction just differs the fast reaction kinetics and the fast transition from soluble Li_2S_n ($n \leq 4$) to insoluble $\text{Li}_2\text{S}_2/\text{Li}_2\text{S}$ enabled by Se-doping, and therefore the balancing of the equation is not required.

4. In the paper, the higher diffusion coefficient of Li^+ is associated with the catalytic effect of Se. The Li^+ diffusion is related to mass transport whereas Se seems to have an effect on the transfer reaction at the interface. Some explanations must be given to

explain clearly the link between the catalytic effect of Se on the charge transfer kinetics (electron) and the diffusion of lithium ions in the electrolyte.

Reply to the Reviewer: The literatures show that metal sulfide with better conductivity than metal oxide deliver better electrochemical performance, resulting from the reduced energy barrier of Li^+ diffusion in the electrode materials and electrolyte during lithiation and delithiation process (*Nature Communications*, 2016, 7, 11203). Moreover, the conductivity of Se_xSPAN is higher than that of SPAN show that the introduction of catalytic Se can effectively accelerate the electron transfer and improve rate performance dramatically. It demonstrates that the introduction of catalytic Se can effectively accelerate the electron transfer and the diffusion of Li^+ ions in the electrode materials during lithiation and delithiation process.

Furthermore, we still try our best to build a simplified model molecules to explore the Li ion diffusion barrier for SPAN and Se_xSPAN using density functional theory (DFT) calculation via VASP code. As a result, we find that the diffusion barrier for Se_xSPAN is indeed smaller than that for SPAN. This result is consistent with the experimental results showing that D_{Li^+} of Se_xSPAN is higher than that of SPAN. This finding likely explain that Se_xSPAN have better reaction kinetics compared with SPAN. A lower barrier can lead to an increase in the diffusion rate according to the exponential rule, and faster diffusion on the Se_xSPAN can promote the reaction of soluble intermediates (Li_2S_n , $n \leq 4$). It suggests that the catalytic amount of Se promotes the diffusion of lithium ions.

We have added the results of DFT calculation in Supplementary Figure 16 and discussed in manuscript. The fast reaction kinetic of Se_xSPAN enabled by Se-doping has been explained more clearly. Hence, the soluble polysulfides can be transformed effectively, which inhibit polysulfides dissolution and the shuttling effect.

Page 15-16: To validate the above-mentioned points, we simulate the diffusion barriers for Li ion on SPAN and Se_xSPAN using density functional theory (DFT) calculation via VASP code. The energy profiles along the diffusion coordinate for Se_xSPAN is 0.39 eV, which is smaller than the diffusion barrier for SPAN, which is 1.33 eV according to our simulations (Supplementary Fig. 16). This result is consistent with the experimental results showing that D_{Li^+} of Se_xSPAN is higher than that of SPAN. This finding likely explain that Se_xSPAN have better reaction kinetics compared with SPAN. A lower barrier can lead to an increase in the diffusion rate according to the exponential rule, and faster diffusion on the Se_xSPAN can promote the reaction of soluble intermediates (Li_2S_n , $n \leq 4$).

Page 20: **DFT calculations.** We perform the density functional theory (DFT) calculation via VASP code. The exchange and correlation energy is in the form of generalized gradient approximation (GGA) with Perdew–Burke–Ernzerhof (PBE). Projector augmented wave (PAW) method is used to depict the electron-ion

interactions. The cutoff energy of 500 eV and Monkhorst–Pack k-meshes of $3 \times 1 \times 1$ are set for the calculations. Vacuum layers of at least 10 Å is needed for non-periodic directions. The Gaussian broadening with a width of 0.05 eV is used for the integration of the first Brillouin zone. The structure is optimized until the Hellmann–Feynman force is smaller than 0.05 eV/Å. Climbing-image nudged elastic band (CI-NEB) method is used to find the Li diffusion barriers.

Supplementary Figure 16. a) SPAN b) Se_xSPAN model for Li ion diffusion barriers employing DFT calculation; c) Energy profiles for diffusion processes of Li ion on SPAN and Se_xSPAN ; Schematic representations of corresponding diffusion pathway for SPAN from d) to e); Schematic representations of corresponding diffusion pathway for Se_xSPAN from f) to g).

5. Why the amount of electrolyte is not the same for carbonate based electrolyte (60 microL) and for the ether one (30 microL), as the PS dissolution is largely associated with its concentration, the comparison is only pertinent with the same amount of electrolyte.

Reply to the Reviewer: Thanks the referee for your valuable comments. It is significant to use the same amount of electrolyte to compare the electrochemical performance in different electrolyte. However, we found that it is necessary to optimize and control the amount of electrolyte. In the case of ether electrolyte, the PS dissolution is indeed associated with the amount of ether electrolyte. Using more ether-based electrolyte will lead to more dissolution and diffusion of polysulfides, resulting in decay of specific capacity. Therefore, the amount of ether-based electrolyte must be the same when comparing Se_xSPAN with SPAN cathode. The different amounts of ether-based electrolyte have been used to test the electrochemical performance. In the coin cell setup, our results suggest that 30 μL ether electrolyte (less than the amount of carbonate)

can lead to good and repeatable electrochemical performance of Se_xSPAN cathode. However, the SPAN cathode still delivered poor electrochemical performance and rapid decay (in Supplementary Figure 7). Meanwhile, if good performance can be achieved, it is always beneficial to use as less electrolyte as possible to lower the cost and improve batteries' energy density. According to our results of Li metal anode after cycling, the Li metal cycling in the ether-based electrolyte is smoother (Supplementary Fig. 11). Nevertheless, our ether electrolyte loading is still much higher than commercial lithium-ion batteries. Hence, in large cell setting, there is still room to keep lowering the amount of electrolyte, which will be beneficial for less polysulfides dissolution, lower cost and higher energy density.

However, in the case of carbonate electrolyte, 60 μL is necessary to obtain better electrochemical performance, which is reasonable considering that the carbonate-based electrolyte could be consumed by the reaction with polysulfides intermediates as well as lithium metal anode. But polysulfides dissolution is not sensitive to the higher amount of carbonate electrolyte used presumably because the polysulfides intermediate could react with carbonate based electrolyte to form a protective layer to prevent further polysulfides (*Advanced Materials*, 2015, 27, 569-575; *Energy Storage Materials*, 2018, 15, 299-307).

6. *The potential of the cell is very low (the average potential is only 1.8V), compared to the classical Li/S cell, some comments on the mechanism must be given in order to explain the potential shape.*

Reply to the Reviewer: Thanks the referee for the valuable comments. Although the structure and the reaction mechanism of SPAN are still not fully clear, many previous work and a most recent one all suggest that sulfur exists as $-\text{S}_x-$ ($2 \leq x \leq 4$) unites in SPAN composite (*Energy Storage Materials*, 2018, 14, 272-278; *Energy Storage Materials*, 2018, 15, 53-64). Such structure is quite different from both elemental sulfur (S_8) and small sulfur molecules (S_{2-4}) in microporous carbon. Therefore quite different reaction intermediates are involved as described in the introduction. The typical elemental sulfur (S_8) will involve a series of polysulfides ($\text{Li}_2\text{S}_{2-8}$) intermediates during electrochemical reaction and show the potential platform at 2.3 V and 2.1 V. On the other hand, the typical small sulfur molecules with the "solid-solid" electrochemical reactions from S_2 to $\text{Li}_2\text{S}_2/\text{Li}_2\text{S}$ in the micro pore show a single reduction peak at 1.7 V in the CV curves, which is related to the reduction of small sulfur molecules confined within the micro porous carbon (*Journal of the American Chemical Society*, 2012, 134, 18510-18513; *ACS NANO*, 2104, 8, 9295-9303).

However, the CV profile of SPAN cathode is much different from typical carbon@sulfur cathode. Experiments with SPAN in ether confirmed the existents and dissolution of soluble polysulfides. Furthermore, it has been found that the CV profile of SPAN shows similarity with that of Li_2S_3 and the peaks in the CV profile of SPAN are higher than the two peaks in that of Li_2S_3 (*Journal of the American Chemical Society*, 2012,

134, 18510-18513; ACS NANO, 2104, 8, 9295-9303). Thus, it also may explain that the SPAN cathode delivers poor cycling in ether-based electrolyte due to higher solubility of higher order polysulfides. Additionally, the Se_xSPAN is very similar to SPAN cathodes which somewhat fall in between elemental sulfur and small molecule sulfur cathodes. Because of the small sulfur units ($-\text{S}_x-$, $2 \leq x \leq 4$) and only lower order soluble polysulfides are involved, the reduction peaks at voltages below 2.1 V rise from the fast reduction of soluble Li_2S_n ($n \leq 4$) and then to insoluble $\text{Li}_2\text{S}_2/\text{Li}_2\text{S}$, which lead to two similar peaks as seen in elemental sulfur but with lower voltage. Thus, the 1.8 V average potential is observed for Se_xSPAN , which also fall in between elemental sulfur and small molecule sulfur cathodes.

Supplementary Figure 15. Comparison of a) the CV curves and b) voltage profiles at 0.1 C of Se_xSPAN , Pure S/C and S/microC.

The comments of the mechanism have been concluded as best as we can. We try our best to explain the lower potential and CV peaks than typical Li-S cell. The explanation has been added in manuscript and Supplementary Figure 15 has been added in the supporting information.

Page 14-15: “It is worth noting that D_{Li^+} for the reduction and oxidation peaks of the $\text{Se}_{0.06}\text{SPAN}$ battery are both higher than the SPAN battery, suggesting that selenium doping facilitates fast Li-ion transport. This finding can be contributed to the catalytic effect of Se on the electrochemical performance for lithium storage, which is consistent of the superior rate performance and cycle performance of Li- Se_xSPAN batteries. A comparison of reduction potentials is also studied to further analyze the effect of Se (Supplementary Fig. 15a). The reduction peaks for Se_xSPAN electrode are 2.07 and 1.77 V, lower than that of pure sulfur electrode (2.32 and 2.02 V), but higher than that of S/microC electrode (1.68 V). The two peaks at 2.32 and 2.02 V correspond to the reduction of S_8 to higher order polysulfides (Li_2S_n , $4 < n < 8$) and then to lower order polysulfides (Li_2S_n , $n \leq 4$)^{27, 50}. The 1.68 V peak is related to the reduction of smaller sulfur molecules confined within the microporous carbon (from S_2 to $\text{Li}_2\text{S}_2/\text{Li}_2\text{S}$)^{25, 27}. Because of the unique sulfur structure ($-\text{S}_x-$, $2 \leq x \leq 4$) in SPAN³⁵, the 2.07 and 1.77 V peaks correspond to the fast reduction of soluble Li_2S_n ($n \leq 4$) and then to insoluble

$\text{Li}_2\text{S}_2/\text{Li}_2\text{S}$. The typical reduction peaks of Se_xSPAN , pure sulfur and S/microC are all in agreement with their own galvanostatic discharge curves (Supplementary Fig. 15b). The 1.8 V average plateaus is observed for Se_xSPAN , which also fall in between elemental sulfur and small molecule sulfur cathodes. It also suggests that only soluble and lower order of polysulfides (Li_2S_n , $n \leq 4$) are involved. Thus the CV curves and voltage profiles suggest a reaction mechanism involving transition between soluble Li_2S_n ($n \leq 4$) and insoluble $\text{Li}_2\text{S}_2/\text{Li}_2\text{S}$ during the lithiation and delithiation process. Owing to the fast Li-ion diffusion and fast reaction kinetics enabled by Se-doping, the soluble polysulfides can be transformed effectively, which inhibit polysulfides dissolution and the shuttling effect. Fig. 5b showed discharge/charge voltage profiles of SPAN and $\text{Se}_{0.06}\text{SPAN}$ electrode at 0.2 A g^{-1} . The $\text{Se}_{0.06}\text{SPAN}$ electrode had a discharge capacity of 1240 mAh g^{-1} , much higher than that of SPAN electrode.”

Reviewers' comments:

Reviewer #1 (Remarks to the Author):

In the manuscript entitled "Ether Compatible Sulfurized Polyacrylonitrile Cathode with Excellent Performance Enabled by Fast Kinetics via Se-doping" the authors have tried to address the initial comments satisfactorily. However, it is anticipated that the following comments should also be addressed prior to the publication of this manuscript in Nature Communication.

Comments regarding the manuscript are listed below:

1. The authors have mentioned in the experimental section that selenium and sulfur powders are ball milled in the ratio of 15:1, 12:1 and 10:1 while in the reviewer's comments the authors are considering S:Se rather than Se:S. It is quite confusing and needs correction accordingly. Also, why the Se ratio is increasing in final SexSPAN samples although it is decreasing in each initial ratio? (As 15:1 sample represented by Se_{0.06}SPAN, so 12:1 and 10:1 will be Se_{0.09}SPAN and Se_{0.14}SPAN, respectively).
2. By comparing Fig. 4(a) and supplementary Fig. 9(a), one can see that the capacity of Se_{0.09}SPAN increased slightly at 1 A/g but the authors claimed that the capacity decreases slightly (Page 12/Line-264). Also, the authors should provide cycling performance comparison in Fig. 9(a) at the same current density.
3. Page 18/Line 385- "...Excess amount of SexS is used..." the author should consider mentioning here whether excess amount is by wt. % or something else.
4. Throughout the manuscript the author is using the word SexS alloy. Then why is it Se-doping in the manuscript title?
5. In the Supplementary Figure 11(c), the separator color is yellowish, but in general this yellowish color appears in ether-based electrolyte and not with carbonate-based electrolyte. This seems to be a contradiction. Please give a reasonable explanation.
6. In Fig. 8 (supplementary information), the authors have compared the cycle performance of the Se_{0.06}SPAN cathodes in ether-based electrolyte with and without LiNO₃ and SPAN cathode with LiNO₃. Following points need more detailed explanation:
 - a) Figure doesn't provide uniform comparison. For e.g. comparison is done at different current density and also cycle number are not same throughout.
 - b) Also, the Se_{0.06}SPAN without LiNO₃ and SPAN with LiNO₃ are showing capacity fading while Se_{0.06}SPAN with LiNO₃ exhibits stable performance. Therefore, stable cyclic performance is only because of LiNO₃ and only advantage with Se-doping is slightly improved capacity. Then how Se doping is beneficial for faster kinetics and hence, the stability?
7. Page 13/Line 268- "The figure also shows that...." the author should consider including the Figure number in this sentence.
8. Page 13/Line 278- "The cathode with 2.3 mg cm⁻² loading cycling in ether-based electrolyte is shown in Supplementary Fig. 9a." The typo error should be corrected in this sentence. It is Fig. 10(a) (supplementary data) and not Fig. 9(a).

Reviewer #2 (Remarks to the Author):

Authors have made good responses and suitable revisions according to the suggestions of previous reviewers. Suggest to accept the manuscript.

Abstract seems too long and the words "This work shows the first ether compatible sulfur cathode" in the line 41/42 and "however only in carbonate electrolyte.30" in the line 77 are not correct. There was already report about SPAN in ether electrolyte, please see the reference, A new ether-based electrolyte for lithium sulfur batteries using a S@pPAN cathode, Chem. Commun., 2018,54, 5478-5481.

Reviewer #3 (Remarks to the Author):

All the corrects required are performed, and the article is largely improved.

No modification is required.

Itemized list of response to the editor' and reviewers' remarks

(Black italic: Reviewer's remarks; Blue type: Our response)

Reviewer(s)' Comments to Author:

Reviewer #1:

In the manuscript entitled "Ether Compatible Sulfurized Polyacrylonitrile Cathode with Excellent Performance Enabled by Fast Kinetics via Se-doping" the authors have tried to address the initial comments satisfactorily. However, it is anticipated that the following comments should also addressed prior to the publication of this manuscript in Nature Communication.

Reply to the Reviewer: Thank you for these useful comments and suggestions. It gives us a great opportunity to explain our work more clearly. From your constructive comments, there are several points that need to be addressed. According to your nice suggestions, we have made extensive corrections. The detailed corrections are listed below.

1. The author have mentioned in the experimental section that selenium and sulfur powders are ball milled in the ratio of 15:1, 12:1 and 10:1 while in the reviewer's comments the authors are considering S:Se rather than Se:S. It is quite confusing and needs correction accordingly.

Also, why Se ratio is increasing in final Se_xSPAN samples although it is decreasing in each initial ratio? (As 15:1 sample represented by $Se_{0.06}SPAN$, so 12:1 and 10:1 will be $Se_{0.09}SPAN$ and $Se_{0.14}SPAN$, respectively).

Reply to the Reviewer: Thanks the referee for the valuable comments. We are sorry for the mistake in our manuscript. It has been modified in the experimental section. The initial mixing ratio of S:Se is 15:1, 12:1 and 10:1 respectively. Since Se:S ratio is increasing as in the Se_xS , it is reasonable that the Se ratio is increasing in final samples ($Se_{0.06}SPAN$, $Se_{0.09}SPAN$ and $Se_{0.14}SPAN$ corresponding to initial mixing weight ratio of S : Se = 15 : 1, 12 : 1 and 10: 1 respectively).

The experiment section of synthesis of Se_xS has been revised with more detailed in manuscript as listed in the following. It is now clear about our experimental section.

Page 17: "Commercial sulfur and selenium powders are ball-milled at a weight ratio of 15:1, 12:1 and 10:1 with ethanol as the dispersant. The resulting mixture is dried at 60 °C for 6 h in a vacuum oven. "

2. By comparing Fig. 4(a) and supplementary Fig. 9(a), one can see that the capacity of $Se_{0.09}SPAN$ increased slightly at 1 A/g but the authors claimed that the capacity decreases slightly (Page 12/Line-264). Also, the authors should provide cycling performance comparison in Fig. 9(a) at the same current density.

Reply to the Reviewer: We have conducted extensive experiments to investigate the electrochemical performance of $\text{Se}_{0.06}\text{SPAN}$ cathode since it has the highest theoretical capacity. The theoretical capacity of $\text{Se}_{0.09}\text{SPAN}$ and $\text{Se}_{0.14}\text{SPAN}$ are lower due to higher proportion of Se. We claim that the capacity decreases slightly as Se loading increases mainly based on the capacity data from the rate performance which is quite easy to obtain, highly repeatable and presumably more conclusive. We show the data in the average level for each sample to make fair comparison. To make it clear, we have added comparison data of rate performance in supplementary Fig 9c.

Nevertheless, the long cycling data is quite difficult to get and take a quite some time. We do have multiple repeating data of $\text{Se}_{0.06}\text{SPAN}$ cathode at 2 A g^{-1} . However, due to the unavoidable inconsistency, limited scale in the lab set up and variation during long cycling runs, the variation of capacity data in long cycling runs are larger than that of rate performance. To be more conservative on our most representative sample, we chose to show long cycling data with below average capacity as represented in Figure 4a. On the other hand, since we only run 200 cycles on $\text{Se}_{0.09}\text{SPAN}$ and $\text{Se}_{0.14}\text{SPAN}$ with only a few examples, the capacity data are probably above the average. Thus, the capacity data just happens to be almost the same or slightly higher than that of $\text{Se}_{0.06}\text{SPAN}$ in the longer cycling run under 1 and 2 A g^{-1} respectively. We show the cycling data of $\text{Se}_{0.09}\text{SPAN}$ and $\text{Se}_{0.14}\text{SPAN}$ mainly to prove the stable cycling performance. We have modified the cycle performance with the same current density in supplementary Fig 9a-b and added rate performance comparison in supplementary Fig 9c.

Supplementary Figure 9. a) Cycle performance of $\text{Se}_{0.09}\text{SPAN}$ and $\text{Se}_{0.14}\text{SPAN}$ cathodes at 1 A g^{-1} . b) Cycle performance of $\text{Se}_{0.09}\text{SPAN}$ and $\text{Se}_{0.14}\text{SPAN}$ cathodes at 2 A g^{-1} . c) Rate performance of Se_xSPAN composite cathodes at various current densities. d) Cycle performance of $\text{Se}_{0.06}\text{SPAN}$ composite cathode with 3 mg cm^{-2} based on the mass of S and Se in the ether-based electrolyte.

Cycle performances of $\text{Se}_{0.09}\text{SPAN}$ and $\text{Se}_{0.14}\text{SPAN}$ cathodes at 1 A g^{-1} and 2 A g^{-1} have been added in supplementary Fig 9a-b. Rate performance of Se_xSPAN composite cathodes at various current densities have been added in supplementary Fig 9c to explain that the specific capacity of different Se_xSPAN cathodes.

Page12: The $\text{Se}_{0.09}\text{SPAN}$ and $\text{Se}_{0.14}\text{SPAN}$ composite cathodes also show good cycle performance (Supplementary Fig. 9a-b) but slightly lower capacity due to higher proportion of Se (Supplementary Fig. 9c).

3. Page 18/Line 385- "...Excess amount of Se_xS is used..." the author should consider mentioning here whether excess amount is by wt. % or something else.

Reply to the Reviewer: According to your suggestions, we have modified the experiment section and highlighted the excess amount is by weight.

Page 18: "A Se_xS and PAN mixture is milled uniformly at a weight ratio of 3:1 for 0.5 h in a mortar and then annealed at $300 \text{ }^\circ\text{C}$ for 2.5 h in argon atmosphere. During this step, excess amount of Se_xS by weight is used. A part of Se and S will be removed from the reactant either in S or Se gas form or H_2S gas form generated in dehydrogenation reaction at $300 \text{ }^\circ\text{C}$."

4. Throughout the manuscript the author is using the word Se_xS alloy. Then why is it Se-doping in the manuscript title?

Reply to the Reviewer: We did use the expression alloy once in our manuscript, which may not be the most appropriate expression. To avoid the confusion, we remove the word "alloy" and use the simple chemical formula. As to the doping concept, we adopt the terminology and concept of Se-doping from a related Se-doped cathode reported previously (*Advanced Materials*, 2017, 29, 1701294). From our procedure and characterization, it is clear to see that the Se_xS materials are prepared at $260 \text{ }^\circ\text{C}$ to form Se-S bond according to the literature procedure (*Energy & Environmental Science*, 2015, 8, 3181-3186). After heat-treatment at 300°C , the Se_xS reacts with PAN to form C-S bond and the Se-S bond still exists. This process is very similar to the preparation of SPAN, in which sulfur exists as $-\text{S}_x^{2-}$ ($2 \leq x \leq 4$) units and covalently bonds to the carbon backbones (*Advanced Materials*, 2002, 14, 963-965; *Advanced Materials*, 2015, 27, 569-575; *Energy Storage Materials*, 2018, 14, 272-278). So in the Se_xSPAN material, presumably there is no Se_xS anymore, but a Se containing SPAN material. Thus Se indeed is doped into the SPAN composite and achieves uniform distribution through Se-S bonding. Therefore, we adopt the terminology and name $\text{Se}_{0.06}\text{SPAN}$, $\text{Se}_{0.09}\text{SPAN}$ and $\text{Se}_{0.14}\text{SPAN}$ as Se-doped SPAN cathodes.

We have modified our manuscript and get rid of expression of alloy.

Page7. "It suggests that there is no $\text{Se}_{0.06}\text{S}$ material in $\text{Se}_{0.06}\text{SPAN}$ composite."

5. In the Supplementary Figure 11(c), the separator color is yellowish, but in general this yellowish color appears in ether-based electrolyte and not with carbonate-based electrolyte. This seems to be a contradiction. Please give a reasonable explanation.

Reply to the Reviewer: Owing to the fast kinetics of $\text{Se}_{0.06}\text{SPAN}$ cathode during cycle, there are barely shuttle effect resulting from the dissolution and diffusion of polysulfides. Thus, the separator used in ether electrolyte is almost colorless like initial separator. However, the separator used in carbonate electrolyte turns yellowish after cycling, which may result from the decomposition of carbonate electrolyte (*Chemical Communications*, 2018, 54, 2288-2291) or the decomposition of LiPF_6 (*ACS Energy Letter.*, 2018, 3, 2921–2930).

6. In Fig. 8 (supplementary information), the authors have compared the cycle performance of the $\text{Se}_{0.06}\text{SPAN}$ cathodes in ether-based electrolyte with and without LiNO_3 and SPAN cathode with LiNO_3 . Following points need more detailed explanation:

a) Figure doesn't provide uniform comparison. For e.g. comparison is done at different current density and also cycle number are not same throughout.

b) Also, the $\text{Se}_{0.06}\text{SPAN}$ without LiNO_3 and SPAN with LiNO_3 are showing capacity fading while $\text{Se}_{0.06}\text{SPAN}$ with LiNO_3 exhibits stable performance. Therefore, stable cyclic performance is only because of LiNO_3 and only advantage with Se-doping is slightly improved capacity. Then how Se doping is beneficial for faster kinetics and hence, the stability?

Reply to the Reviewer: Thanks for the valuable comments. It gives us another chance to explain this important issue more clearly. We have added more data and explanation in the following.

(a) We have added cycling performance of SPAN with LiNO_3 under 0.4 A g^{-1} in Figure 8a, which shows inferior cycling performance that that under 0.3 A g^{-1} . We also keep cycle numbers to be 70 to compare these data. Nevertheless both our data and data from previous literature suggest that in general, traditional SPAN cathode exhibits severe shuttle effect and poor cycling in ether-based electrolyte with or without LiNO_3 (*Journal of the American Chemical Society*, 2015, 137, 12143-12152).

(b) Our data have shown that the structure of $\text{Se}_{0.06}\text{SPAN}$ is very similar to SPAN. So if the stable cycling performance is only because of LiNO_3 , SPAN should show good cycling data with LiNO_3 like case of $\text{Se}_{0.06}\text{SPAN}$ with LiNO_3 . Meanwhile, even without LiNO_3 , the cycling performance of $\text{Se}_{0.06}\text{SPAN}$ is clear better than SPAN. So LiNO_3 is not the determining factor for the cathode cycling performance, but rather useful for good cycling of Lithium metal anode. In general, Capacity fading in Li-S batteries could result from two main resources: 1) S cathode; 2) Li metal anode. Since previous reports have shown that LiNO_3 can react with Li metal in ether electrolyte to form SEI to protect Li metal anode to a certain degree. So cycling with LiNO_3 can avoid anode fading problem and allows us to evaluate S cathode performance. Therefore the control experiments in SPAN in ether electrolyte with and without

LiNO₃ demonstrate that in the case of SPAN, the capacity fading is mainly associated with SPAN cathode. It is the stable cycling is only due to the presence of LiNO₃, then SPAN should show good cycling results. But it is not the case. On the contrary, the fact that Se_{0.06}SPAN with LiNO₃ can cycle very well in ether electrolyte then proves that Se_{0.06}SPAN cathode doesn't show cathode-related capacity fading. Both experimental and theoretical results conclude that it is due to the fast redox conversion in Se_{0.06}SPAN enabled by Se-doping. Meanwhile, the fast kinetics will also lead to significant enhancement of rate performance (much higher capacity under the same current density, especially high current density) which has been shown in Figure 3, Supplementary data Figure 8b and Figure 13.

To furthermore clarify the cycling performance, we also design an experiment to prove that the LiNO₃ additive in ether electrolyte mainly has an effect on the protection of Li metal anode to achieve better electrochemical performance. Firstly, the Li-Se_{0.06}SPAN cell is cycled in ether-based electrolyte with LiNO₃ for 10 cycles to form stable SEI of Li metal anode. Then, the cell is unpacked and reassembled using ether-based electrolyte without LiNO₃. The cycle performance of Se_{0.06}SPAN cathode is still good after the LiNO₃ is removed from the electrolyte (Supplementary Fig. 8c-d). Since SPAN couldn't cycle well even with LiNO₃, we didn't run the same experiment on SPAN. The collective results further show that the superior electrochemical performance, especially the stable cycling results of Se_{0.06}SPAN cathode in ether electrolyte is only caused by the introduction of small amount of Se.

Supplementary Figure 8. a) Cycle performance of the Se_xSPAN and SPAN cathodes using ether electrolyte with and without LiNO₃. b) Rate performance of SPAN in the ether-based electrolyte. c) Cycle performance of the Se_xSPAN cathode using ether-based electrolyte, firstly cycled for 10 cycles with LiNO₃ and subsequently

cycled without LiNO₃. d) The corresponding electrochemical discharge and charge profiles of Se_{0.06}SPAN at various cycles.

We have modified Supplementary Figure 8 and added the cycling results of the Se_{0.06}SPAN cathodes without LiNO₃ in ether-based electrolyte to explain the effect of catalytic Se more clearly.

Page 12: “The phenomenon for traditional SPAN cathode in ether-based electrolyte can also be seen in other literature reports. From our control experiment by first cycling the Li-Se_{0.06}SPAN cell for 10 cycles with LiNO₃, then removing the electrolyte, reassembling the electrodes and cycling in ether-based electrolyte without LiNO₃. The cycle performance of Se_{0.06}SPAN cathode is still very stable after LiNO₃ is removed (Supplementary Fig. 8c-d). It suggests that the superior electrochemical performance of Se_{0.06}SPAN cathode is only caused by the introduction of small amount Se. The LiNO₃ additive presumably only has an effect on the protection of Li metal anode for better cycling performance. The high reversible capacity, long cycle life and high Coulombic efficiency of Se_{0.06}SPAN composites cathodes demonstrate that the Se can accelerate redox kinetics and prevent the dissolution of the lithium polysulfides effectively.”

7. Page 13/Line 268- “The figure also shows that.....” the author should consider including the Figure number in this sentence.

Reply to the Reviewer: Thanks for the suggestion. The Figure number has been added in the sentence to be more clearly.

Page 12: “It is clearly shown that the cycle performances of Se_{0.06}SPAN composite cathode are higher than the representative values of literature reports (seen in Supplementary Table 7 and Fig. 13b). The Fig. 4c also shows that based on the mass of the composite, Se_{0.06}SPAN composites deliver a capacity of 546 mAh g⁻¹ at second cycle, corresponding to an excellent utilization ratio (75%) to its theoretical capacity (726 mAh g⁻¹), and 416 mAh g⁻¹ even after 800 cycles based on the Se_{0.06}SPAN composite. Compared to literature reports, the electrochemical performances of Se_{0.06}SPAN composite cathode are at the top level (Supplementary Table 6 and Table 7, Fig. 13).”

8. Page 13/Line 278- “The cathode with 2.3 mg cm⁻² loading cycling in ether-based electrolyte is shown in Supplementary Fig. 9a.” The typo error should be corrected in this sentence. It is Fig. 10(a) (supplementary data) and not Fig. 9(a).

Reply to the Reviewer: Thanks for pointing it out. We are sorry for the mistake in our manuscript. The typo error has been modified in the sentence.

Page 13: “The cathode with 2.3 mg cm⁻² loading cycling in ether-based electrolyte is shown in Supplementary Fig. 10a. After 50 cycles, the cycled CR2025 coin cells in two different electrolytes are unpacked.”

Reviewer #2:

Reviewer #2 (Remarks to the Author):

Authors have made good responses and suitable revisions according to the suggestions of previous reviewers. Suggest to accept the manuscript.

Reply to the Reviewer: We would like to thank you first for positive comments of our manuscript. According to your nice suggestions, we have made corrections.

Abstract seems too long and the words "This work shows the first ether compatible sulfur cathode" in the line 41/42 and "however only in carbonate electrolyte.30" in the line 77 are not correct. There was already report about SPAN in ether electrolyte, please see the reference, A new ether-based electrolyte for lithium sulfur batteries using a S@pPAN cathode, Chem. Commun., 2018,54, 5478-5481.

Reply to the Reviewer: Thank the referee for the constructive comments. As suggested, we have shortened the abstract and change the expression in the line 41/42 to "usually in carbonate electrolyte".

Page 1-2: "Traditional sulfurized polyacrylonitrile (SPAN) are suggested to contain S_n ($n \leq 4$) and show good electrochemical results usually in carbonate electrolytes. However inferior results in ether electrolytes suggest that the solubility Li_2S_n ($n \leq 4$) is high enough to trump the limited redox conversion and leads to the dissolution and the shuttling effect. Herein, we introduce small amount of selenium doping in sulfurized polyacrylonitrile to accelerate the redox conversion of Li_2S_n ($n \leq 4$). The designed Se_xSPAN ($x < 0.15$, $\sim 50\text{wt}\%$ Se_xS) cathodes deliver excellent electrochemical performance in both carbonate and ether electrolytes, showing a high reversible capacity up to 1300 mAh g^{-1} at 0.2 A g^{-1} (0.13 C), excellent active material utilization ratio (84%) and extremely high rate with capacity up to 900 mAh g^{-1} at 10 A g^{-1} (6.5 C). The $Li-Se_xSPAN$ cells can cycle up to 800 cycles with nearly 100 % Coulombic efficiency and 0.029% capacity decay per cycle. The dissolution and diffusion of polysulfide are successfully suppressed owing to fast reaction kinetics of Se_xSPAN . This work shows an ether compatible sulfur cathode involving intermediate Li_2S_n ($n \leq 4$), demonstrates one of the best rate and cycling performance of sulfur cathodes at the same time and provides a promising solution towards applicable lithium sulfur batteries."

Page 3: "Surprisingly, sulfurized polyacrylonitrile (SPAN), firstly reported by Wang and co-workers in 2002, seems to be a promising cathode which exhibits good capacity, reasonable rate capability, nearly 100% Columbic efficiency, good cycling performance and presumably no polysulfides dissolution usually in carbonate electrolyte."

Reviewer #3 (Remarks to the Author):

*All the corrects required are performed, and the article is largely improved.
No modification is required.*

Reply to the Reviewer: Thank the referee for the positive comments.